# Towards Formalizing Spuriousness of Biased Datasets Using Partial Information Decomposition

**Barproda Halder**                                                    *bhalder@umd.edu*
*Department of Electrical and Computer Engineering*
*University of Maryland, College Park*

**Faisal Hamman**                                                      *fhamman@umd.edu*
*Department of Electrical and Computer Engineering*
*University of Maryland, College Park*

**Pasan Dissanayake**                                                  *pasand@umd.edu*
*Department of Electrical and Computer Engineering*
*University of Maryland, College Park*

**Qiuyi Zhang**                                                        *qiuyiz@google.com*
*Google Research*

**Ilia Sucholutsky**                                                   *is2961@princeton.edu*
*Department of Computer Science*
*Princeton University*

**Sanghamitra Dutta**                                                  *sanghamd@umd.edu*
*Department of Electrical and Computer Engineering*
*University of Maryland, College Park*

**Reviewed on OpenReview:** `https://openreview.net/forum?id=zw6UAPYmyx`

## Abstract

Spuriousness arises when there is an association between two or more variables in a dataset that are not causally related. In this work, we propose an explainability framework to preemptively disentangle the nature of such spurious associations in a dataset before model training. We leverage a body of work in information theory called Partial Information Decomposition (PID) to decompose the total information about the target into four non-negative quantities, namely *unique information (in core and spurious features, respectively), redundant information, and synergistic information.* Our framework helps anticipate when the core or spurious feature is indispensable, when either suffices, and when both are jointly needed for an optimal classifier trained on the dataset. Next, we leverage this decomposition to propose a novel measure of the spuriousness of a dataset. We arrive at this measure systematically by examining several candidate measures, and demonstrating what they capture and miss through intuitive canonical examples and counterexamples. Our framework *Spurious Disentangler* consists of segmentation, dimensionality reduction, and estimation modules, with capabilities to specifically handle high-dimensional image data efficiently. Finally, we also perform empirical evaluation to demonstrate the trends of unique, redundant, and synergistic information, as well as our proposed spuriousness measure across 6 benchmark datasets under various experimental settings. We observe an agreement between our preemptive measure of dataset spuriousness and post-training model generalization metrics such as worst-group accuracy, further supporting our proposition. The code is available at `https://github.com/Barproda/spuriousness-disentangler`.

# 1 Introduction

The success of machine learning is heavily determined by the quality of datasets used for training or fine-tuning. Spurious patterns (Haig, 2003) arise when two or more variables are associated in a dataset even though they do not have a causal relation, e.g., image classifiers on Waterbird dataset (Wah et al., 2011) learn to use the background rather than the foreground for classification, because most waterbirds are photographed on a water background (see Fig. 1). This pattern in the dataset misleads a classifier into learning an undesirable spurious link between the target label (bird type) and background ("spurious" feature) as opposed to the foreground (core feature). Spuriousness in datasets may result in deceptively high performance on in-distribution datasets but significantly hinders generalization on out-of-distribution datasets, e.g., accuracy on minority groups like waterbirds with land background is low (Lynch et al., 2023; Sagawa et al., 2019; Puli et al., 2023).

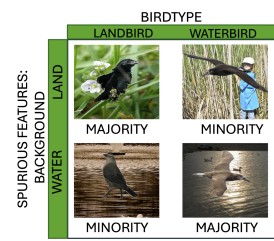

Figure 1: Spuriousness in Waterbird dataset due to sampling bias.

Despite advances in dataset-based and model-training-based approaches to mitigate such spurious patterns (Ye et al., 2024; Srivastava, 2023; Ghouse et al., 2024), the notion of spuriousness in any given dataset has classically lacked a formal definition. To address this gap, in this work, we ask the following question: *Given a dataset and a split of core and spurious features, can we preemptively quantify the spuriousness of the dataset before training?* In essence, our goal is to arrive at a framework that would help anticipate the feature preferences of an optimal classifier prior to training.

To this end, we provide an information-theoretic explainability framework to disentangle the nature of spurious associations in a dataset, i.e., how the information about the target variable is distributed among the spurious and core features. We leverage a body of work in information theory called Partial Information Decomposition (PID) (Bertschinger et al., 2014; Banerjee et al., 2018), which has its roots in statistical decision theory. We note that classical information-theoretic measures such as mutual information (Cover & Thomas, 2012) capture the entire statistical dependency between two random variables but fail to capture how this dependency is distributed among those variables, i.e., the structure of the multivariate information. Partial Information Decomposition (PID) addresses this nuanced issue by providing a formal way of *disentangling* the joint information content between the core and spurious features into *unique, redundant, or synergistic information*. We leverage this decomposition to systematically arrive at a novel measure of dataset spuriousness with empirical evaluation on high-dimensional image datasets. This work provides a more nuanced understanding of the interplay between spurious and core features in a dataset that can better inform dataset quality assessment. Our main contributions can be summarized as follows:

**Unraveling nature of spurious associations leveraging PID:** We leverage PID to disentangle the total information about a target ($Y$) in the core ($F$) and spurious ($B$) features into four non-negative terms: *unique information (in core and spurious features respectively), redundant information, and synergistic information* (see Proposition 1). We elucidate four types of statistical dependencies captured by the PID terms (see Fig. 3), providing preemptive insights on when an optimal classifier might find a spurious feature more informative or useful than the core features. We establish how unique information quantifies the informativeness of a feature over another for predicting $Y$ (see Theorem 1). Then, redundant information turns out to be the common information that can be obtained from either the spurious or core features, allowing a classifier to potentially choose either without any preference. An interesting term is synergy that captures scenarios when both spurious and core features are jointly informative about the target $Y$ but not individually (classifier likely to use both spurious and core).

**Novel measure of dataset spuriousness:** Though many works attempt to prevent a model from learning spurious patterns, there is limited theoretical understanding of how to quantify the spuriousness of a dataset given a choice of core and spurious features. In this work, we leverage PID to propose a novel measure of the undesirable spuriousness of a dataset ($M_{sp}$) that steers predictors into choosing the spurious features over the core (see Proposition 2). We arrive at this measure systematically by examining several candidate measures, and demonstrating what they capture and miss through intuitive canonical examples and counterexamples. Our measure provides a fundamental understanding of feature informativeness for a classification task, enabling dataset quality assessment and interpretability.

**Spuriousness Disentangler:** We propose an autoencoder-based explainability framework that we call – Spuriousness Disentangler – to obtain the four PID values as well as our spuriousness measure $M_{sp}$ for high-dimensional image data. The framework consists of three modules: (i) Segmentation: This module performs segmentation to separate the foreground (core features $F$) and background (spurious features $B$) for every image, either using pre-trained semantic segmentation models (Lin et al., 2017) or CLIPSeg (Lüddecke & Ecker, 2022), which is an Open-Vocabulary Semantic Segmentation model if necessary; (ii) Dimensionality Reduction: An autoencoder converts high-dimensional images into lower-dimensional, discrete feature representations. The dimensionality reduction and clustering are performed jointly through minimization of a joint loss function, drawing inspiration from Guo et al. (2017). (iii) Estimation: The final step includes the estimation of the joint probability distribution of the acquired lower-dimensional representation, followed by optimization (James et al., 2018; Liang et al., 2023) to compute PID values and $M_{sp}$.

**Empirical results:** Since our proposed framework is a preemptive dataset explainability framework, the goal of our experiments is to show broad agreement between our anticipations from the dataset before training and the post-training behavior of the models for various experimental setups. We examine four experimental setups: i) Both core and spurious features are available; (ii) Either core or spurious is available; (iii) Segmentation to obtain core and spurious features; and (iv) Non-spatial spuriousness. Our evaluation spans six datasets: Waterbird (Wah et al., 2011), Adult (Becker & Kohavi, 1996), CelebA (Lee et al., 2020), Dominoes (Shah et al., 2020), Spawrious (Lynch et al., 2023), and Colored MNIST (Arjovsky et al., 2019). We observe a negative correlation between our proposed measure of dataset spuriousness $M_{sp}$ and post-training model generalization metrics, such as the worst-group accuracy for each experimental setting. We also study Grad-CAM (Selvaraju et al., 2017) visualizations and intersection-over-union (IoU) metric (Rezatofighi et al., 2019) to further confirm which features are actually being emphasized by the model.

A framework for dataset explainability provides an alternative to combating spuriousness during training by providing preemptive insights to inform the training process (analogous to "nutrition labels" Yang et al. (2018) or "datasheets for datasets" Gebru et al. (2021)). By enabling dataset quality check and cleansing prior to training, it can bypass expensive adversarial training, often used to avoid spurious patterns. Having clean datasets for fine-tuning is particularly valuable in the era of large foundation models when one has limited control over the training process.

**Related Works:** There are several perspectives on spurious correlation (see Haig (2003); Kirichenko et al. (2022); Izmailov et al. (2022); Wu et al. (2023); Ye et al. (2023); Liu et al. (2023); Stromberg et al. (2024); Singla & Feizi (2021); Moayeri et al. (2023); Lynch et al. (2023) and the references therein; also see surveys Ye et al. (2024); Srivastava (2023); Ghouse et al. (2024)). Spuriousness mitigation techniques are broadly divided into two groups: (i) Dataset-based techniques (Goel et al., 2020; Kirichenko et al., 2022; Wu et al., 2023; Moayeri et al., 2023; Liu et al., 2021) and (ii) Learning-based techniques (Liu et al., 2023; Yang et al., 2023; Ye et al., 2023; Zhang et al., 2022). Among dataset-based techniques, Kirichenko et al. (2022) shows that last-layer fine-tuning of a pre-trained model with a group-balanced subset of data is sufficient to mitigate spuriousness. Wu et al. (2023) proposes a concept-aware spurious correlation mitigation technique. There are also some works that try to separate spurious and core features in the feature space of deep neural networks using external feedback (Sohoni et al., 2020; Kattakinda et al., 2022). Recent work Wang & Wang (2024) looks into the problem through the mathematical lens of separability of the spurious and core features under a mixture of Gaussian assumptions (also assuming a split between core and spurious). Ye et al. (2023) discusses how the noise in the core feature plays a role in a model's reliance on it. Our novelty lies in investigating the problem of spurious patterns through the lens of PID, rooted in statistical decision theory, focusing on quantifying the spuriousness of a dataset for interpretability and quality assessment. Our work isolates four specific types of statistical dependencies in the dataset, providing a more nuanced understanding (see Fig. 3) going beyond solely identifying a model's reliance on a specific feature.

Partial Information Decomposition (PID) (Williams & Beer, 2010; Bertschinger et al., 2014) is an active area of research, beginning to be used in different domains of neuroscience and machine learning (Tax et al., 2017; Dutta et al., 2020; Hamman & Dutta, 2024; Ehrlich et al., 2022; Liang et al., 2024; Wollstadt et al., 2023; Mohamadi et al., 2023; Venkatesh et al., 2024; Dutta et al., 2021; Dissanayake et al., 2024). We also refer to a survey (Dutta & Hamman, 2023). However, interpreting spuriousness in datasets through the lens of PID is unexplored. Additionally, there is limited work on calculating PID values for high-dimensional multivariate

continuous data. Some existing works (Dutta et al., 2021; Venkatesh et al., 2024) handle continuous data with Gaussian assumptions while (Pakman et al., 2021) considers one-dimensional multivariate case. Hence, estimating PID for high-dimensional data through proper dimensionality reduction and discretization is also fairly open. For dimensionality reduction, different learning based methods exist (Hotelling, 1933; Law & Jain, 2006; Lee & Verleysen, 2005; Wang et al., 2015; 2014; Sadeghi & Armanfard, 2023). Similarly, for discretization, different clustering algorithms exist, e.g., k-means clustering (MacQueen et al., 1967; Bradley et al., 2000), deep embedded clustering (Xie et al., 2016). In this work, we train an autoencoder to jointly learn a good lower-dimensional representation of the input image data in a self-supervised manner (with additional bottleneck structure) while also clustering simultaneously to deal with the challenge of high-dimensional real-valued image data.

## 2 Preliminaries

Let $X = (X_1, X_2, \ldots, X_d)$ be the random variable denoting the input (e.g., an image) where each $X_i \in \mathcal{X}$ denotes a finite set of values that each feature can take. The core features (e.g., the foreground) will be denoted by $F \subseteq X$, and the spurious features (e.g., the background) will be denoted by $B = X \backslash F$. We typically use the notation $\mathcal{B}$ and $\mathcal{F}$ to denote the range of values for the spurious and core features. Let $Y$ denote the target random variable, e.g., the true labels which lie in the set $\mathcal{Y}$, and the model predictions are given by $\hat{Y} = f_\theta(X)$ (parameterized by $\theta$). Generally, we use the notation $P_A$ to denote the distribution of random variable $A$, and $P_{A|B}$ to denote the conditional distribution of random variable $A$ conditioned on $B$. Depending on the context, we also use more than one random variable as subscript, e.g., $P_{ABY}$ denotes the joint distribution of $(A, B, Y)$. Whenever necessary, we also use the notation $Q_A$ to denote an alternate distribution on the random variable $A$ that is different from $P_A$. We also use the notation $P_{A|B} \circ P_{B|C}$ to denote a composition of two conditional distributions given by: $P_{A|B} \circ P_{B|C}(a|c) = \sum_{b \in \mathcal{B}} P_{A|B}(a|b) P_{B|C}(b|c) \ \forall a \in \mathcal{A}, \ c \in \mathcal{C}$, where $\mathcal{A}$, $\mathcal{B}$, and $\mathcal{C}$ denote the range of values that can be taken by random variables $A$, $B$, and $C$.

**Background on PID:** We provide a brief background on PID that would be relevant for the rest of the paper (also see Fig. 2). The classical information-theoretic quantification of the total information that two random variables $A$ and $B$ together hold about $Y$ is given by mutual information $\mathrm{I}(Y; A, B)$ (see (Cover & Thomas, 2012) for a background on mutual information). Mutual information $\mathrm{I}(Y; A, B)$ is defined as the KL divergence (Cover & Thomas, 2012) between the joint distribution $P_{YAB}$ and the product of the marginal distributions $P_Y \otimes P_{AB}$ and would go to zero if and only if $(A, B)$ is independent of $Y$. *Intuitively, this mutual information captures the total predictive power about $Y$ that is present jointly in $(A, B)$ together, i.e., how well one can learn $Y$ from $(A, B)$ together.* However, $\mathrm{I}(Y; A, B)$ only captures the total information content about $Y$ jointly in $(A, B)$ and does not unravel what is unique or shared between $A$ and $B$.

PID (Bertschinger et al., 2014) provides a mathematical framework that decomposes the total information content $\mathrm{I}(Y; A, B)$ into four non-negative terms:

$$\mathrm{I}(Y; A, B) = \mathrm{Uni}(Y{:}B|A) + \mathrm{Uni}(Y{:}A|B)$$
$$+ \mathrm{Red}(Y{:}A, B) + \mathrm{Syn}(Y{:}A, B).$$

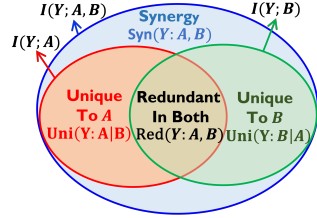

Figure 2: $\mathrm{I}(Y; A, B)$ is decomposed into four non-negative terms.

Here, $\mathrm{Uni}(Y{:}A|B)$ denotes the *unique information* about $Y$ that is only in $A$ but not in $B$ and $\mathrm{Uni}(Y{:}B|A)$ denotes the *unique information* about $Y$ that is only in $B$ but not in $A$. Next, $\mathrm{Red}(Y{:}A, B)$ denotes redundant information (common knowledge) about $Y$ in both $A$ and $B$. Lastly, $\mathrm{Syn}(Y{:}A, B)$ is an interesting term that denotes the synergistic information that is present only jointly in $A, B$ but not in any one of them individually, e.g., a public and private key can jointly reveal information not in any of them alone.

*Example to Understand PID:* Let $Z = (Z_1, Z_2, Z_3)$ with each $Z_i \sim$ i.i.d. Bern(1/2). Let $A = (Z_1, Z_2, Z_3 \oplus N)$, $B = (Z_2, N)$, and $N \sim$ Bern(1/2) which is independent of $Z$. Here, $\mathrm{I}(Z; A, B) = 3$ bits. The unique information about $Z$ that is contained only in $A$ and not in $B$ is effectively in $Z_1$. Thus, $\mathrm{Uni}(Z{:}A|B) = \mathrm{I}(Z; Z_1) = 1$ bit. Redundant information about $Z$ that is contained in both $A$ and $B$ is effectively in $Z_2$ and is given by $\mathrm{Red}(Z{:}A, B) = \mathrm{I}(Z; Z_2) = 1$ bit. Synergistic information about $Z$ that is not contained in either

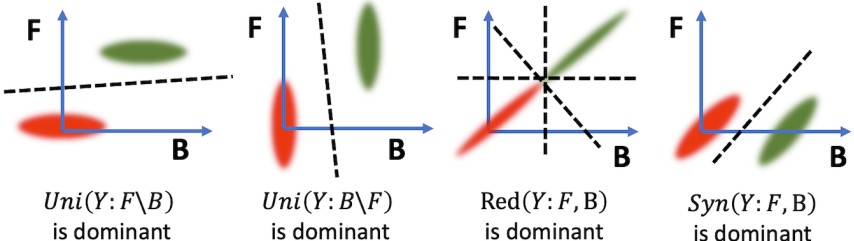

$Uni(Y{:}F{\backslash}B)$ is dominant  $\quad$  $Uni(Y{:}B{\backslash}F)$ is dominant  $\quad$  $Red(Y{:}F, B)$ is dominant  $\quad$  $Syn(Y{:}F, B)$ is dominant

Figure 3: Canonical examples distilling four types of statistical dependencies involving core and spurious features when any one PID term is dominant and its effect on the Bayes optimal classifier. In the first two cases, unique information in either $F$ or $B$ is dominant, and they are indispensable to the optimal classifier. When redundant information is dominant, the optimal classifier can pick either $F$ or $B$ without preference. The fourth scenario is interesting, where $B$ is independent of the label $Y$, and yet it contributes to the optimal classifier along with $F$.

$A$ or $B$ alone, but is contained in both of them together is effectively in the tuple $(Z_3 \oplus N, N)$, and is given by $\mathrm{Syn}(Z{:}A, B)=\mathrm{I}(Z; (Z_3 \oplus N, N)) = 1$ bit. This accounts for the 3 bits in $\mathrm{I}(Z; A, B)$.

Defining any one of the PID terms suffices for obtaining the others. This is because of another relationship among the PID terms as follows (Bertschinger et al., 2014): $\mathrm{I}(Y; A) = \mathrm{Uni}(Y{:}A|B)+\mathrm{Red}(Y{:}A, B)$. Essentially $\mathrm{Red}(Y{:}A, B)$ is viewed as the sub-volume between $\mathrm{I}(Y; A)$ and $\mathrm{I}(Y; B)$ (see Fig. 2). Hence, $\mathrm{Red}(Y{:}A, B) = \mathrm{I}(Y; A) - \mathrm{Uni}(Y{:}A|B)$. Lastly, $\mathrm{Syn}(Y{:}A, B) = \mathrm{I}(Y; A, B) - \mathrm{Uni}(Y{:}A|B) - \mathrm{Uni}(Y{:}B|A) - \mathrm{Red}(Y{:}A, B)$ (can be obtained once both unique and redundant information has been obtained). Here, we include a popular definition of $\mathrm{Uni}(Y{:}A|B)$ from (Bertschinger et al., 2014) which is computable using convex optimization.

**Definition 1** (Unique Information (Bertschinger et al., 2014))**.** *Let $\Delta$ be the set of all joint distributions on $(Y, A, B)$ and $\Delta_P$ be the set of joint distributions with same marginals on $(Y, A)$ and $(Y, B)$ as the true distribution $P_{YAB}$, i.e., $\Delta_P = \{Q_{YAB} \in \Delta: Q_{YA} = P_{YA} \text{ and } Q_{YB} = P_{YB}\}$. Then,*

$$\mathrm{Uni}(Y{:}A|B) = \min_{Q \in \Delta_P} \mathrm{I}_Q(Y; A|B). \tag{1}$$

*Here $\mathrm{I}_Q(Y; A|B)$ denotes the conditional mutual information when $(Y, A, B)$ have joint distribution $Q_{YAB}$ rather than $P_{YAB}$.*

## 3 Theoretical Contributions

### 3.1 Unraveling the nature of spurious associations with PID

**Proposition 1** (Proposed Disentanglement)**.** *For a given data distribution, the total predictive power of the spurious features $B$ and core features $F$ about the target variable $Y$ can be decomposed into four non-negative components:*

$$\mathrm{I}(Y; F, B) = \mathrm{Uni}(Y{:}B|F) + \mathrm{Uni}(Y{:}F|B) + \mathrm{Red}(Y{:}F, B) + \mathrm{Syn}(Y{:}F, B).$$

For each term in Proposition 1, we now explain their nuanced role for any given dataset.

*Interpreting Unique Information* $\mathrm{Uni}(Y{:}B|F)$ *and* $\mathrm{Uni}(Y{:}F|B)$*:* Unique information captures information that is unique in one feature and cannot be obtained from another. To explain the role of unique information in interpreting spuriousness, we draw upon a concept in statistical decision theory called Blackwell Sufficiency (Blackwell, 1953) which investigates when a random variable is "more informative" (or "less noisy") than another for inference (also relates to stochastic degradation of channels (Venkatesh et al., 2023; Raginsky, 2011)). Let us first discuss this notion intuitively when trying to infer $Y$ using two random variables $F$ and $B$. Suppose there exists a transformation on $F$ to give a new random variable $B'$ which is always equivalent to $B$ for predicting $Y$ (similar predictive power). We note that $B'$ and $B$ do not necessarily

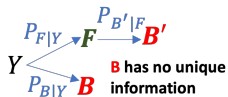

Figure 4: Blackwell Sufficiency

have to be the same since we only care about inferring $Y$. In fact, $B$ and $B'$ can have additional irrelevant information that do not pertain to $Y$, but solely for the purpose of inferring $Y$, they need to be equivalent. Then, $F$ will be regarded as "sufficient" with respect to $B$ for predicting $Y$ since $F$ can itself provide all the information that $B$ has about $Y$ (see Fig. 4 and first two cases of Fig. 3).

**Definition 2** (Blackwell Sufficiency (Blackwell, 1953)). *A conditional distribution $P_{F|Y}$ is Blackwell sufficient with respect to another conditional distribution $P_{B|Y}$ if and only if there exists a stochastic transformation (equivalently another conditional distribution $P_{B'|F}$ with both $B$ and $B' \in \mathcal{B}$) such that $P_{B'|F} \circ P_{F|Y} = P_{B|Y}$.*

In fact, the unique information $\text{Uni}(Y{:}B|F)$ is 0 if and only if $P_{F|Y}$ is Blackwell sufficient with respect to $P_{B|Y}$ (see Theorem 1, the proof is given in the Appendix C).

**Theorem 1** (Interpretability Insights from Unique Information). *The following properties hold:*

- $\text{Uni}(Y{:}B|F) \leq \text{I}(Y;B)$ *and goes to 0 if the spurious feature $B$ is independent of the target $Y$. However,* $\text{Uni}(Y{:}B|F)$ *may be 0 even if* $\text{I}(Y;B) > 0$.

- $\text{Uni}(Y{:}B|F) = 0$ ***if and only if*** $P_{F|Y}$ *is Blackwell sufficient with respect to $P_{B|Y}$.*

- $\text{Uni}(Y{:}B|F) \leq \text{Uni}(Y{:}B'|F')$, *i.e., it is non-decreasing if some features from the core set are moved to the spurious set, i.e., $B' = B \cup W$ and $F' = F \backslash W$.*

Since unique information $\text{Uni}(Y{:}B|F) = 0$ if and only if $P_{F|Y}$ is Blackwell Sufficient with respect to $P_{B|Y}$, we note that $\text{Uni}(Y{:}B|F) > 0$ captures the "departure" from Blackwell Sufficiency, and thus quantifies *relative informativeness. Intuitively, what this means is that for a data distribution, there is no such transformation on core feature $F$ that is equivalent to the spurious feature $B$ for the purpose of predicting $Y$. This essentially makes spurious feature $B$ indispensable for predicting $Y$, forcing a model to emphasize it in decision-making.* A similar argument can be made for $\text{Uni}(Y{:}F|B)$. Furthermore, $\text{Uni}(Y{:}B|F)$ also satisfies an intuitive property that as more features get categorized as spurious instead of core, the unique information in the spurious set would keep increasing.

*Interpreting Redundant Information* $\text{Red}(Y{:}F, B)$: Redundant information about the target variable $Y$ is the information that can be obtained from either the spurious features $B$ or the core features $F$ without any preference towards either. We consider the following canonical example to interpret the role of redundant information $\text{Red}(Y{:}F, B)$ for predicting the target variable $Y$ (third case of Fig. 3).

**Lemma 1** (Redundancy). *Let $B = Y + N_B, F = Y + N_F$ where noise $N_B$ and $N_F$ are Gaussian such that $N_B = N_F = N \sim \mathcal{N}(0, \sigma_N^2)$ and $N \perp\!\!\!\perp Y$. In this case, (i) an optimal predictor $\hat{Y}$ can either utilize $B$ or $F$ with neither being indispensable, i.e., $\hat{Y} = f(B)$ or $f(F)$ or $f(B, F)$; and (ii) $B$ and $F$ will only have redundant information with the other PID terms being 0.*

*Interpreting Synergistic Information:* Synergistic information $\text{Syn}(Y{:}F, B)$ is an interesting term that emerges when spurious features $B$ and core features $F$ together reveal more about the target variable $Y$ than what can be revealed by either of them alone. In essence, it is the "extra" or "emergent" information that arises only when multiple features interact, rather than when they are considered separately.

**Lemma 2** (Synergy). *Let $B=N$, $F=Y+N$ where $Y \sim Bern(1/2)$, $N \sim \mathcal{N}(0, \sigma_N^2)$, $N \perp\!\!\!\perp Y$ and $\sigma_N^2 \gg 1$. Then, (i) an optimal predictor $\hat{Y} = f(F, B) = F - B$ (uses both $F$ and $B$); and (ii) $\text{I}(Y;B)$ and $\text{I}(Y;F) \approx 0$ but $\text{I}(Y;B, F)$ is still significant due to synergistic information $\text{Syn}(Y{:}B, F)$.*

For this example (fourth case in Fig. 3), both $F$ and $B$ alone will have limited predictive power when $N$ has high variance. However, using $F$ and $B$ together, one can perfectly predict $Y$, e.g., an optimal predictor is $\hat{Y} = f(F, B) = F - B$. Here $\text{I}(Y;B) = 0$, and we also show that $\text{I}(Y;F) \approx 0$ (see Lemma 8 in Appendix C). However, the synergistic information $\text{Syn}(Y{:}F, B)$ is still significant. Since $\text{I}(Y;F) \approx 0$, we contend that here $B$ essentially denoises the core feature $F$, enhancing its predictive power. Thus, synergistic information captures an interesting nuanced interplay between core and spurious, not captured by the other PID terms.

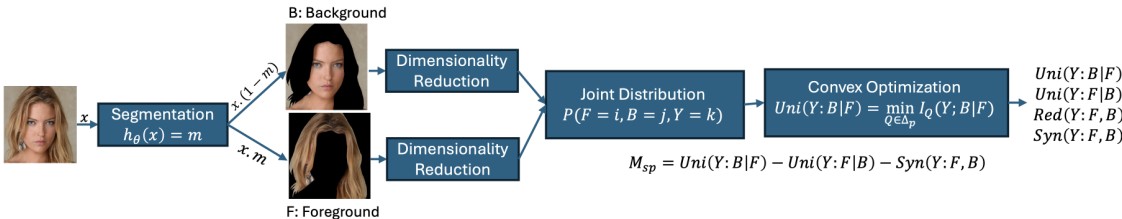

Figure 5: Spuriousness Disentangler: An autoencoder-based explainability framework to handle high dimensional continuous image data with 3 modules: (i) Segmentation of images into background (spurious features) and foreground (core features); (ii) Dimensionality reduction involving an autoencoder with bottleneck and clustering; and (iii) Estimation of the joint distribution followed by the computation of PID values through convex optimization and computing $M_{sp}$.

### 3.2 A novel measure of dataset spuriousness

Our objective is to quantify a dataset's spuriousness, which steers machine learning models towards the spurious features over the core features. To this end, we will examine some candidate measures ($M_{sp}$) of spuriousness through examples and counterexamples and systematically arrive at a measure that meets our requirements. Since we are trying to capture spuriousness which arises when the target variable $Y$ is associated with the spurious features $B$, we might first consider the mutual information $\text{I}(Y;B)$ as a candidate measure for spuriousness since it captures the dependence between $Y$ and $B$.

**Candidate Measure 1.** $M_{sp} = \text{I}(Y;B)$.

**Counterexample 1.** We refer to the example in Lemma 1 where $\text{Uni}(Y{:}B|F) = 0$. Hence, $\text{I}(Y;B) = \text{Uni}(Y{:}B|F) + \text{Red}(Y{:}F,B) = \text{Red}(Y{:}F,B)$. Here, our candidate measure $M_{sp} = \text{I}(Y;B)$ is positive, which would indicate "spuriousness," i.e., undesirable steering towards $B$. However, in this case, the model can use either spurious features $B$ or core features $F$ (see Lemma 1) without any preference. Thus, $\text{I}(Y;B)$ is not well suited to be a measure of undesirable spuriousness.

Since redundant information can lead to the utilization of either spurious or core features, another candidate measure of spuriousness might be obtained by subtracting the desirable dependence $\text{I}(Y;F)$ from the undesirable dependence $\text{I}(Y;B)$, i.e., $M_{sp} = \text{I}(Y;B) - \text{I}(Y;F)$. For the example in Lemma 1, this new $M_{sp} = 0$, indicating no preference towards spurious or core features.

**Lemma 3.** *Let $B = Y + N_B, F = Y + N_F$ where noise $N_B$ and $N_F$ are standard Gaussian noises with $N_B \sim \mathcal{N}(0, \sigma^2_{N_B})$, $N_F \sim \mathcal{N}(0, \sigma^2_{N_F})$ and $N_B \perp\!\!\!\perp Y$, $N_F \perp\!\!\!\perp Y$. Now if $\sigma^2_{N_F} \gg \sigma^2_{N_B}$, (i) the optimal classifier relies strongly on spurious feature $B$; and (ii) $\text{Uni}(Y{:}B|F) > 0$.*

If $\sigma^2_{N_F} \gg \sigma^2_{N_B}$, then $\text{I}(Y;B) > \text{I}(Y;F)$, i.e., $M_{sp} > 0$ (see Lemma 8 in Appendix C). Here, the output of a model is more likely to be $\hat{Y} = f(B)$ and the model might be more prone to utilizing the spurious features $B$ (see Fig. 3). On the other hand, if $\sigma^2_{N_F} \ll \sigma^2_{N_B}$, then $\text{I}(Y;F) > \text{I}(Y;B)$, i.e., $M_{sp} < 0$ . In this case, the output of the model is also more likely to be $\hat{Y} = f(F)$, and the model might lean towards the core features $F$. Hence, $M_{sp} = \text{I}(Y;B) - \text{I}(Y;F)$ might seem like a suitable measure to quantify spuriousness, i.e., steering models towards $B$ over $F$.

**Candidate Measure 2.** $M_{sp} = \text{I}(Y;B) - \text{I}(Y;F) = \text{Uni}(Y{:}B|F) - \text{Uni}(Y{:}F|B)$.

**Counterexample 2.** Consider Lemma 2 where the optimal predictor $\hat{Y} = F - B$ utilizing both the spurious features $B$ and core features $F$. Here, this $M_{sp} \approx 0$ (Lemma 2). However, for this particular example, since the prediction is jointly influenced by both core features $F$ and spurious features $B$, we contend that a measure of spuriousness should not be 0. The measure should therefore include a term that considers the joint contribution of both of these features, capturing the fact that here $B$ simply helps in denoising and enhancing the predictive capabilities of the core features $F$. This aspect is precisely captured by synergistic information $\text{Syn}(Y{:}F,B)$. Hence, we also include it in $M_{sp}$, leading to the following proposed measure.

**Proposition 2** (Measure of Spuriousness $M_{sp}$). *Our proposed measure of spuriousness is given by:*

$$M_{sp} = \text{Uni}(Y{:}B|F) - \text{Uni}(Y{:}F|B) - \text{Syn}(Y{:}F, B). \tag{2}$$

## 4  Methodology: Spuriousness Disentangler

We propose an autoencoder-based explainability framework – that we call Spuriousness Disentangler – to disentangle the PID values and compute the measure $M_{sp}$ (see Fig. 5 and Algorithm 1) for a given dataset. The framework consists of three modules: segmentation, dimensionality reduction, and estimation.

---

**Algorithm 1:** Spuriousness Disentangler: An Autoencoder-Based Explainability Framework

---

**Input** : Encoder input $F$ or $B$, decoder output $F'$ or $B'$, autoencoder parameters $\theta$, cluster centers $\mu_j$ (parameters of the clustering layer), embedded point $z_i$ (output of the clustering layer), soft label $q_i$ (output of the clustering layer), hyperparameter $\gamma$, pretrain epochs $e_p$, maximum epochs $e_{\max}$, update interval $T$, batch number $b$, threshold $\delta$, target variable $Y$.

**Step 1: Segmentation**;
Perform segmentation to separate the core features $F$ and spurious features $B$ if necessary;
**Step 2: Dimensionality Reduction (Autoencoder Training)**;
**for** $e = 1$ **to** $e_p$ **do**
    compute $L_r \leftarrow \|F - F'\|_2^2$ ;                // Reconstruction loss (MSE)
    compute gradient $\nabla_\theta L_r$ and update $\theta$ via gradient descent
**end**

Initialize $\mu_j$ performing k-means clustering and $p_{ij} \leftarrow \frac{q_{ij}(0)^2}{\sum_i q_{ij}(0)} / \sum_j \frac{q_{ij}(0)^2}{\sum_i q_{ij}(0)}$;

**for** $e = 1$ **to** $e_{\max}$ **do**
    compute $L \leftarrow L_r + \gamma L_c$;
    where, $L_c \leftarrow KL(P\|Q) = \sum_i \sum_j p_{ij} \log \frac{p_{ij}}{q_{ij}(e)}$ and $q_{ij}(e) \leftarrow \frac{(1+\|z_i(e)-\mu_j(e)\|^2)^{-1}}{\sum_j (1+\|z_i(e)-\mu_j(e)\|^2)^{-1}}$;
    **if** $(b-1) \bmod T == 0$ **and** $(b \neq 1$ **and** $e \neq 1)$ **then**
        update $p_{ij} \leftarrow \frac{q_{ij}(e)^2}{\sum_i q_{ij}(e)} / \sum_j \frac{q_{ij}(e)^2}{\sum_i q_{ij}(e)}$, $\text{preds}_{\text{new}} \leftarrow \arg\max q_{ij}(e)$ and $\delta_{\text{label}} \leftarrow \frac{\sum(\text{preds}_{\text{new}} \neq \text{preds}_{\text{old}})}{|\text{preds}|}$;
        **if** $\delta_{label} < \delta$ **then**
            break;
        **end**
        $\text{preds}_{\text{old}} \leftarrow \text{preds}_{\text{new}}$;
    **end**
    compute gradient $\nabla_\theta L$ and update $\theta$ via gradient descent
**end**
Use discrete values from clusters of latent representations for $F$ and $B$;
**Step 3: Estimation**;
    Estimate $P(F=i, B=j, Y=k)$;             // Joint distribution estimation
    Compute $\text{Uni}(Y{:}F|B) = \min_{Q \in \Delta_P} I_Q(Y; F|B)$ according to Eq. 1;    // Iterative optimization
    Calculate $M_{sp}$ using Eq.2;
**Output:** Measure of spuriousness $M_{sp}$;

---

**Segmentation:** This step involves separating the core ($F$) from the spurious ($B$) features. Publicly available segmentation masks are used where available. For datasets without explicit core or spurious feature information, masks can be generated using pre-trained semantic segmentation models (Lin et al., 2017). Another possibility is to use CLIPSeg (Lüddecke & Ecker, 2022), an Open-Vocabulary Semantic Segmentation model, to automatically isolate various objects in the foreground to approximately obtain the core and spurious features. Experiments for various scenarios are provided in Section 5.

**Dimensionality Reduction:** Since we are dealing with high-dimensional image data, our next module compresses them into lower-dimensional discrete vectors. We propose to use an autoencoder, a deep neural

network consisting of an encoder and a decoder, as shown in Fig. 6 to jointly do dimensionality reduction and clustering. We incorporate a bottleneck structure from (Sadeghi & Armanfard, 2023) with convolutional autoencoders of Guo et al. (2017) to obtain more informative lower-dimensional representation of the input image. Along the lines of Guo et al. (2017), we obtain the clusters of the low-dimensional data $q$ by optimizing a joint loss function defined as $L = L_r + \gamma L_c$ where $L_r$ is the representation loss, $L_c$ is the clustering loss, and $\gamma$ is a non-negative constant. See Appendix C.4 for more details.

**Estimation:** The final step includes the estimation of the joint distribution and the PID values, also leading to the proposed measure $M_{sp}$. The joint distribution is obtained by computing the normalized 3D histogram of the discrete clusters of the foreground, background, and binary target variable. Then, the PID values are estimated from the joint distribution using the DIT package (James et al., 2018), which is a Python package for discrete information theory. We use $I_{BROJA}$ developed in (Bertschinger et al., 2014) to compute PID which solves the convex optimization problem in Definition 1 and results in four non-negative terms, namely, $\mathrm{Uni}(Y{:}B|F)$, $\mathrm{Uni}(Y{:}F|B)$, $\mathrm{Red}(Y{:}F,B)$, and $\mathrm{Syn}(Y{:}F,B)$. We use them to calculate the measure $M_{sp}$. In case of a multiclass classification task (more than two classes), we use the PID estimator proposed by (Liang et al., 2023).

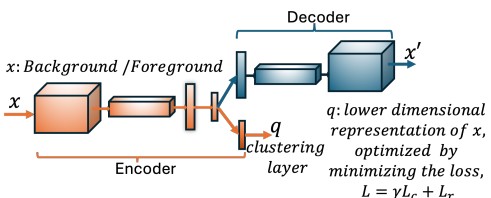

Figure 6: Dimensionality reduction module: Autoencoder with clustering to have discrete lower-dimensional embedding.

## 5 Experimental Results

Since our proposition is a preemptive dataset explainability framework, the objective of our experiments is to see how our anticipations from dataset before training agree with post-training model generalization metrics like worst-group accuracy, SHAP, IoU, etc. In particular, we consider four setups: (i) Both core and spurious features are available; (ii) Either core or spurious is available; (iii) Segmentation to obtain core and spurious features; and (iv) Non-spatial spuriousness. We conduct experiments on six datasets: Waterbird (Wah et al., 2011), Adult (Becker & Kohavi, 1996), CelebA (Lee et al., 2020), Dominoes (Shah et al., 2020), Spawrious (Lynch et al., 2023), and Colored MNIST (Arjovsky et al., 2019). We begin with using our explainability framework, namely Spuriousness Disentangler, on each dataset (often with dataset-specific sampling biases and variations) to compute the PID values and $M_{sp}$. We fine-tune the pre-trained ResNet-50 (He et al., 2016) model and calculate the worst-group accuracy over all groups for the Waterbird, CelebA, Dominoes, Spawrious, and Colored MNIST datasets. For the tabular dataset Adult, we train the XGBoost (Chen & Guestrin, 2016) model and calculate the worst-group accuracy. More details of the experiments are provided in the Appendix D along with a comparison of our proposed measure $M_{sp}$ with other measures in Table 2, Appendix D.2.

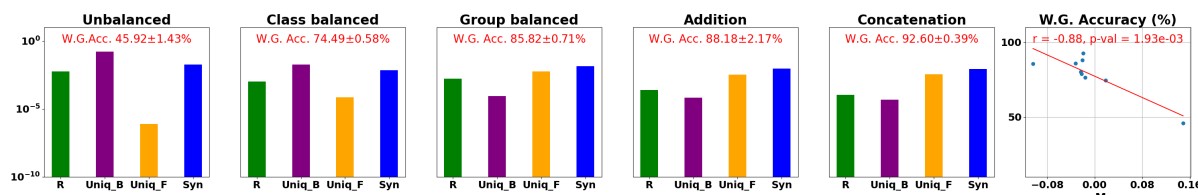

Figure 7: Bar-plot showing the redundant information (R), unique information in background (Uniq-B) and foreground (Uniq-F), and Synergistic information (Syn) for different variants (essentially different sampling biases) of the **Waterbird** dataset. Observe that the Uniq-B decreases and Uniq-F increases for group-balanced, addition, and concatenation datasets compared to those of the unbalanced dataset. Also, observe a negative trend between $M_{sp}$ and W.G. Acc. Note that the y-axis of the first five subplots is in log scale.

**1. Both core and spurious features available:** For the Waterbird, Dominoes, and Adult datasets, core and spurious features are well-defined and accessible. In Waterbird, the bird's pixels serve as core features, while the background is spurious. In the synthetic Dominoes dataset, car or truck images are core, and

digits zero or one are spurious. For the tabular Adult dataset, *gender* is considered spurious, while *age, education-num,* and *hours-per-week* are chosen as core.

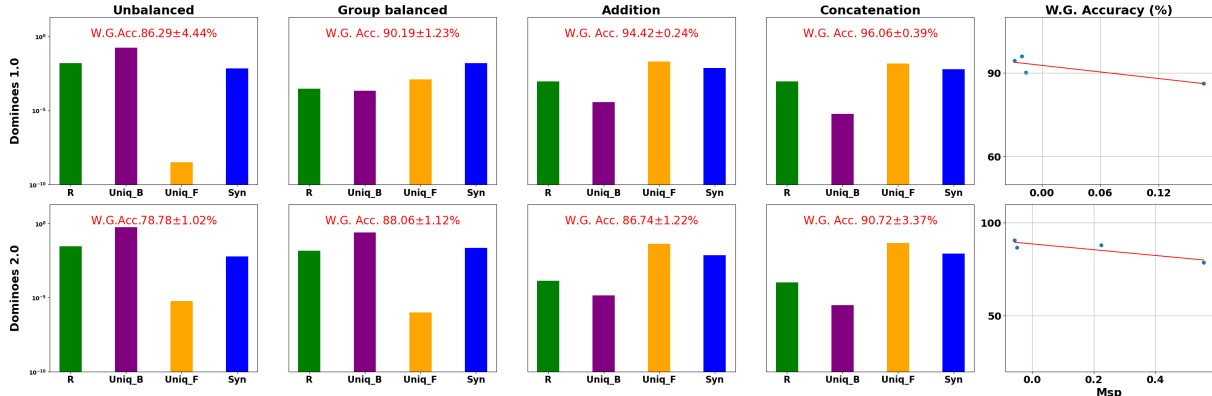

Figure 8: Bar-plot showing the redundant information (R), unique information in background (Uniq-B) and foreground (Uniq-F), and Synergistic information (Syn) for **Dominoes** dataset. The Uniq-B decreases group-balanced and background mixed datasets, and the Uniq-F increases for background mixed datasets compared to the unbalanced dataset. Also observe a negative trend between the $M_{sp}$ and W.G. Acc.

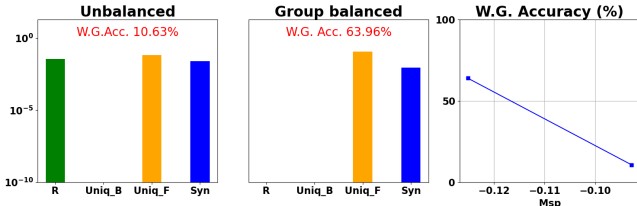

Figure 9: Bar-plot showing the PID values for the tabular dataset: **Adult**. The last plot shows a negative relationship between the worst-group accuracy and the measure of spuriousness $M_{sp}$. Note that the y-axis of the first two subplots is in log scale.

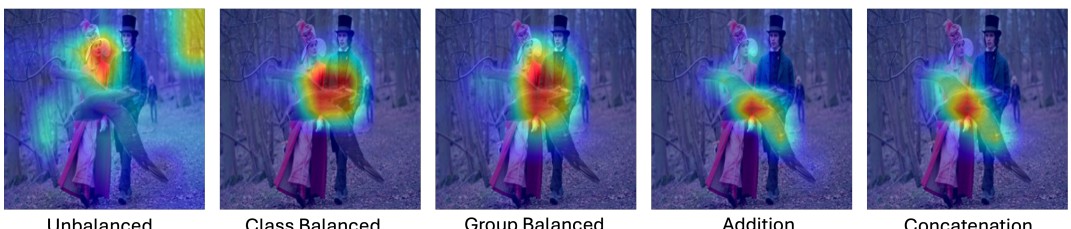

Figure 10: Examples of Grad-CAM images of Waterbird: For the unbalanced dataset, the model adds more emphasis (red regions) to the background, while in the class-balanced, group-balanced, addition, and concatenation versions, the foreground gets more emphasis.

**Observations:** Fig. 7 (Waterbird), Fig. 8 (Dominoes), and Fig. 9 (Adult) show that the unique information in the background decreases and the unique information in the foreground increases when the dataset is modified from unbalanced to other variants. We also observe a negative correlation between the measure of spuriousness $M_{sp}$ and the worst-group accuracy (see the Pearson correlation coefficient ($r$) with corresponding p-value in Fig. 7). *The negative correlation between our proposed dataset spuriousness measure $M_{sp}$ and the model generalization metric worst-group accuracy indicates that $M_{sp}$ can serve as an indicator of dataset quality before training.* Fig. 10 and Fig 18 in Appendix D.4.1 show that when the dataset is balanced or mixed background, the Grad-CAM (Selvaraju et al., 2017) emphasizes more on the core features (the red regions) while in the unbalanced dataset, the background is more emphasized, which results in poor worst-group accuracy. See Table 1 and Table 3 in Appendix D.2, which show how Intersection-over-union (IoU)

between the ground truth masks and Grad-CAM masks changes over different variants of the Waterbird dataset and the variation of PID values for different numbers of clusters, respectively.

**2. Only core features available:** For the CelebA dataset, the core features are the pixels corresponding to *hair*. However, the spurious one is not given directly. We consider everything excluding *hair* as spurious.

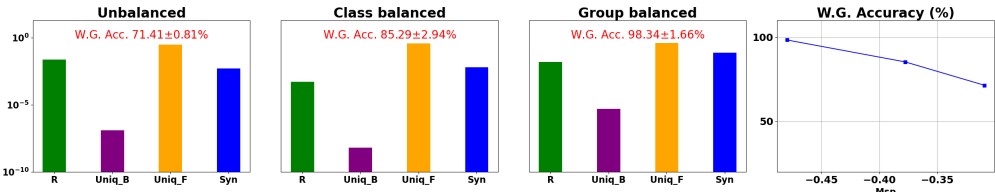

Figure 11: CelebA: Observe that the Uniq-F and Synergy increase for class-balanced and group-balanced datasets compared to that of the unbalanced dataset, and a trade-off between $M_{sp}$ and W.G. Acc.

**Observations**: Fig. 11 shows the PID values for unbalanced, class-balanced, and group-balanced CelebA datasets. *Firstly,* the unique information in the foreground is the most prominent one among all other PID values. Observe that the Uniq-B is almost negligible. The Uniq-F increases while the dataset is class-balanced or group-balanced, along with the increasing worst-group accuracy. There is a negative trend between worst-group accuracy and the measure of spuriousness $M_{sp}$. *Secondly,* the Grad-CAM images (see Fig. 21 in Appendix. D.4.2) show that the model focuses on the hair for the balanced dataset, but for the unbalanced dataset, it emphasizes more on the face.

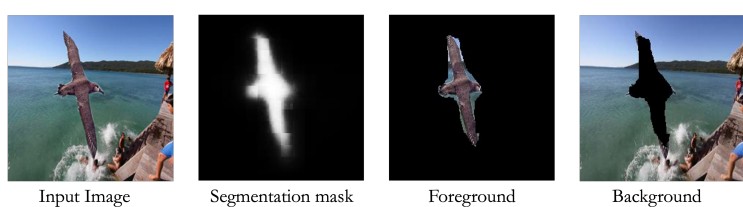

Figure 12: The segmentation mask is obtained by zero-shot image segmentation using CLIPSeg (Lüddecke & Ecker, 2022). We get the foreground by multiplying the input image with the *mask* and background by multiplying $(1 - mask)$.

**3. Segmentation to obtain features:** When explicit information about the core and spurious features is unavailable, we need to perform segmentation. We perform experiments with the Waterbird dataset assuming the unavailability of the segmentation masks. We leverage CLIPSeg (Lüddecke & Ecker, 2022), an Open-Vocabulary Semantic Segmentation (OVSS) model, to generate the mask for the bird object with the prompt "bird" (details in Appendix D.1), thus utilizing only partial knowledge about the target object. The generated masks are applied to the original images to separate the foreground and background (see Fig. 12). Fig. 13a reveals a negative correlation between the worst-group accuracy and increasing values of $M_{sp}$, calculated using the obtained background and foreground.

Alternatively, for the Spawrious dataset, we use a pre-trained segmentation model (Lin et al., 2017) to generate the mask of the dog and separate the foreground "dog" from the background. Fig. 13b shows that the redundancy and unique information in the background decrease while the unique information in the foreground and synergy increase when the dataset is group-balanced. We also observe a negative trend between $M_{sp}$ and the worst-group accuracy, showing the effectiveness of the measure.

**4. Non-spatial spuriousness:** We apply our framework to the Colored MNIST (Arjovsky et al., 2019) dataset, which has non-spatial biases. In this setting, we define the grayscale digit images as the core features $F$ and the colored digit images as the spurious features $B$, leveraging partial knowledge that the target variable should correspond to the actual digits rather than their color. We perform two experiments: (i) a 10-class digit classification and (ii) a binary classification, where digits $< 5$ are class 0, and $\geq 5$ are class 1. There is a negative correlation between the measure of spuriousness $M_{sp}$ and worst-group accuracy (W.G. Acc.) (see the Pearson correlation coefficient ($r$) with corresponding p-value), consistent with observations

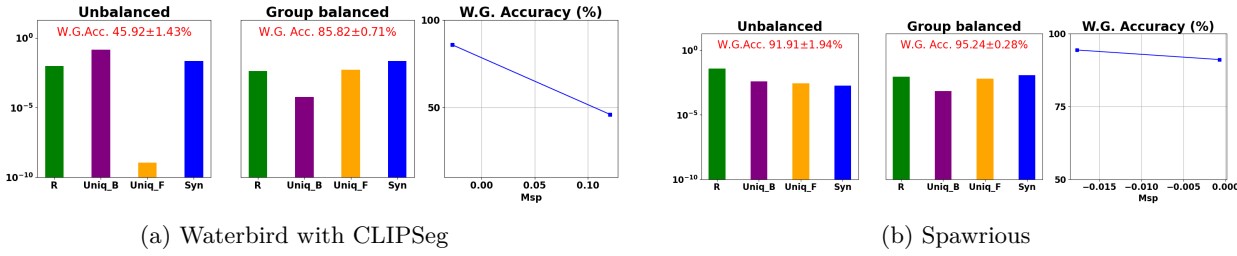

(a) Waterbird with CLIPSeg                    (b) Spawrious

Figure 13: **(a,b)** The first two plots show the change in redundancy, unique information, and the synergistic information. The last plot shows a negative relationship between the W.G. Acc. and the measure of spuriousness $M_{sp}$. Note that the y-axis of the first two subplots is in log scale.

from other datasets, validating the broad applicability of our proposed measure (see Fig. 14). Furthermore, by extending our framework to the multiclass setting, we demonstrate its enhanced versatility.

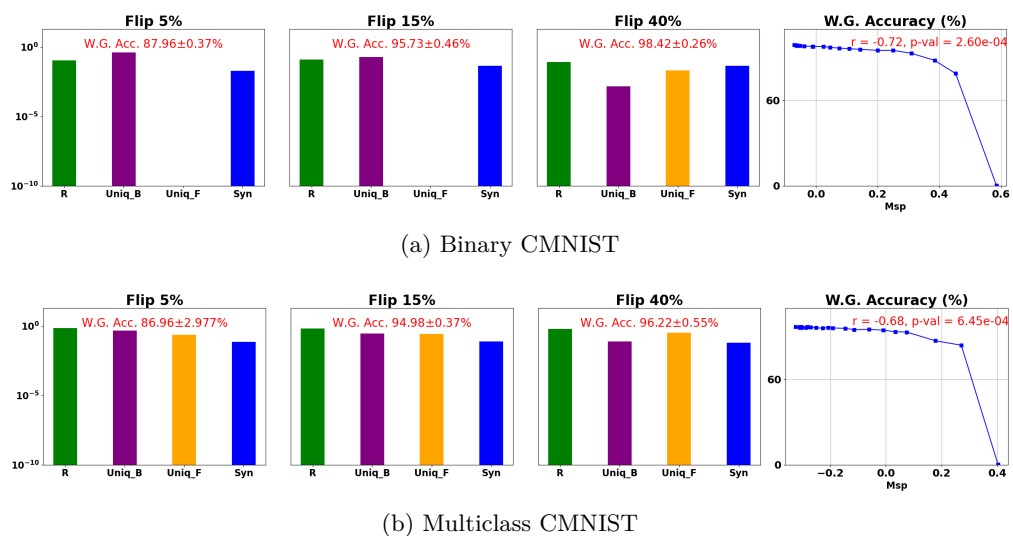

(a) Binary CMNIST

(b) Multiclass CMNIST

Figure 14: **(a,b)** The first three plots show the change in redundancy, unique information, and the synergistic information. The last plot shows a negative relationship between the W.G. Acc.(%) and the measure of spuriousness $M_{sp}$. Note that the y-axis of the first three subplots is in log scale.

## 6 Conclusion

This work brings in a novel perspective on spuriousness by identifying four types of statistical dependencies in a dataset, leveraging the PID framework: when $F$ or $B$ is indispensable (unique information dominant), when either $F$ or $B$ suffices (redundant information dominant), and when both $F$ and $B$ are jointly needed (synergistic information dominant). This leads us to propose a measure of dataset spuriousness as an efficient way to assess dataset quality before performing the actual training or fine-tuning, which can be computationally intensive, particularly in the era of foundational models. We also perform experiments on several datasets to check if our anticipations from data agree with post-training model generalization metrics such as worst-group accuracy. Notably, worst-group accuracy and our measure of spuriousness are not mathematically the same thing: our measures anticipate how the Bayes optimal classifier(s) should behave, and the empirical measures show how specific models actually behave (when trained on that dataset without doing anything specific for spuriousness). Nonetheless, we observe an interesting correlation across a broad range of experimental setups, further validating the efficacy of our measure. Our implementation is able to handle high-dimensional image data and estimate the PID terms (Broader Impacts in Appendix B).

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

## A Limitations

(i) Identifying spurious features and core features of a given dataset automatically is not always straightforward. Future work will look into alternate techniques, such as causal discovery (Zanga et al., 2022) (recently using LLMs (Liu et al., 2024)), as well as validation on NLP datasets. (ii) Separation of spurious and core features often relies on manual annotations, heuristic segmentation, or prior domain knowledge, potentially introducing subjective bias in feature categorization. (iii) The estimation is highly data-dependent. A small change in the dataset can greatly affect the PID values. In fact, estimating information-theoretic quantities such as mutual information is inherently difficult in high-dimensional settings (Belghazi et al., 2018), including formal limits on sample complexity for all estimators (McAllester & Stratos, 2020), and PID estimation would also inherit these challenges. Moreover, PID requires solving an additional optimization problem beyond mutual information computation, which can introduce additional difficulties. Future work will look into estimation error analysis and alternate techniques (Lyu et al., 2024; Goswami & Merkley, 2024; Gomes & Figueiredo, 2024; Mediano et al., 2025). (iv) The efficiency and robustness of the Spurious Disentangler can also be improved. (v) Additionally, there can be groups of spurious features rather than just one, which can have nuanced interplay among them, which is another interesting direction.

## B Broader Impact

Quantifying spuriousness has significant broader impacts across multiple domains. Quantification of dataset spuriousness might improve the trustworthiness of AI in several high-stakes and safety-critical applications, such as healthcare, which can directly impact people's lives. Spurious patterns often lead to biased predictions, particularly in sensitive domains such as hiring, lending, or criminal sentencing. Going beyond existing works, our research paves the way for improved understanding of the nature of spurious relationships, enabling interpretability, which could also have significant implications in auditing.

A framework for dataset explainability provides an alternative to combating spuriousness during training by providing preemptive insights to inform the training process (analogous to "nutrition labels" Yang et al. (2018) or "datasheets for datasets" Gebru et al. (2021)). By enabling dataset quality check and cleansing prior to training, it can bypass expensive adversarial training, often used to avoid spurious patterns. Having clean datasets for fine-tuning is particularly valuable in the era of large foundation models when one has limited control over the training process.

## C Appendix to Theoretical Results

### C.1 Relevant Mathematical Results

PID (Bertschinger et al., 2014; Banerjee et al., 2018) provides a mathematical framework that decomposes the total information content $\mathrm{I}(Y; A, B)$ into four non-negative terms:

$$\mathrm{I}(Y; A, B) = \mathrm{Uni}(Y{:}B|A) + \mathrm{Uni}(Y{:}A|B) + \mathrm{Red}(Y{:}A, B) + \mathrm{Syn}(Y{:}A, B). \tag{3}$$

In addition to this equation, the PID terms also satisfy the following relationships (Bertschinger et al., 2014; Banerjee et al., 2018):

$$\mathrm{I}(Y; A) = \mathrm{Uni}(Y{:}A|B) + \mathrm{Red}(Y{:}A, B). \tag{4}$$

$$\mathrm{I}(Y; A|B) = \mathrm{Uni}(Y{:}A|B) + \mathrm{Syn}(Y{:}A, B). \tag{5}$$

Now, defining any one of the PID terms is sufficient to obtain all four by using these relationships. In this work, we use a popular definition of unique information from (Bertschinger et al., 2014; Banerjee et al., 2018) as defined in Definition 1, which can be computed by solving a convex optimization problem (Bertschinger et al., 2014; Banerjee et al., 2018).

One of the most desirable properties of this definition is that all four PID terms are non-negative.

**Lemma 4** (Nonnegativity of PID)**.** *All four PID terms* $\mathrm{Uni}(Y{:}B|A)$, $\mathrm{Uni}(Y{:}A|B)$, $\mathrm{Red}(Y{:}A, B)$, *and* $\mathrm{Syn}(Y{:}A, B)$ *are nonnegative as per Definition 1.*

This result is proved in Bertschinger et al. (2014, Lemma 5).

**Lemma 5** (Monotonicity under local operations on $B$)**.** *Let $B = f(B')$ where $f(\cdot)$ is a deterministic function. Then, we have:*

$$\mathrm{Uni}(Y{:}B|A) \leq \mathrm{Uni}(Y{:}B'|A).$$

This result is derived in Banerjee et al. (2018, Lemma 31).

**Lemma 6** (Monotonicity under adversarial side information)**.** *For all $(Y, B, A, W)$, we have:*

$$\mathrm{Uni}(Y{:}B|A, W) \leq \mathrm{Uni}(Y{:}B|A).$$

This result is derived in Banerjee et al. (2018, Lemma 32).

**Lemma 7.** $\mathrm{Uni}(Y{:}B|F) = 0$ *if and only if there exists a row-stochastic matrix $T \in [0, 1]^{|\mathcal{F}| \times |\mathcal{B}|}$ such that:* $P_{YB}(Y = y, B = b) = \sum_{f \in \mathcal{F}} P_{YF}(Y = y, F = f)T(f, b)$ *for all $y \in \mathcal{Y}$ and $b \in \mathcal{B}$.*

*Proof.* This result is from Bertschinger et al. (2014). Here, we include a proof for completeness.

If $\mathrm{Uni}(Y{:}B|F) = 0$, then we have: $\min_{Q \in \Delta_P} \mathrm{I}_Q(Y; B|F) = 0$ where $\Delta_P = \{Q \in \Delta : Q_{YF}(Y = y, F = f) = P_{YF}(Y = y, F = f)$ and $Q_{YB}(Y = y, B = b) = P_{YB}(Y = y, B = b)\}$. Thus, there exists a distribution $Q \in \Delta_P$ such that $Y$ and $B$ are independent given $F$ under the joint distribution $Q$. Then, we have

$$P_{YB}(Y = y, B = b) = Q_{YB}(Y = y, B = b) \tag{6}$$

$$= \sum_{f \in \mathcal{F}} Q_{YFB}(Y = y, F = f, B = b) \tag{7}$$

$$= \sum_{f \in \mathcal{F}} Q_{B|YF}(B = b|Y = y, F = f)Q_{YF}(Y = y, F = f) \tag{8}$$

$$\overset{(a)}{=} \sum_{f \in \mathcal{F}} Q_{B|YF}(B = b|Y = y, F = f)P_{YF}(Y = y, F = f) \tag{9}$$

$$\overset{(b)}{=} \sum_{f \in \mathcal{F}} Q_{B|F}(B = b|F = f)P_{YF}(Y = y, F = f) \tag{10}$$

$$\overset{(c)}{=} \sum_{f \in \mathcal{F}} T(f, b)P_{YF}(Y = y, F = f). \tag{11}$$

Here, (a) holds because $P_{YF} = Q_{YF}$ for all $Q \in \Delta_P$, (b) holds because under joint distribution $Q$, variables $Y$ and $B$ are independent given $F$, and (c) simply chooses $T(f, b) = Q_{B|F}(B = b|F = f)$ which is a function of $(f, b)$ and will lead to a row-stochastic matrix $T$ since $\sum_{b \in \mathcal{B}} T(f, b) = \sum_{b \in \mathcal{B}} Q_{B|F}(B = b|F = f) = 1$.

Next, we prove the converse. Suppose, such a row-stochastic matrix $T$ exists such that:

$$P_{YB}(Y = y, B = b) = \sum_{f \in \mathcal{F}} T(f, b)P_{YF}(Y = y, F = f).$$

Now, we can define a joint distribution $Q^*$ such that:

$$Q^*(Y = y, F = f, B = b) = P_{YF}(Y = y, F = f)T(f, b). \tag{12}$$

We can show that $Q^*$ is a valid probability distribution since $T$ is row stochastic.

$$\sum_{y\in\mathcal{Y}}\sum_{b\in\mathcal{B}}\sum_{f\in\mathcal{F}} Q^*(Y=y, F=f, B=b) = \sum_{y\in\mathcal{Y}}\sum_{b\in\mathcal{B}}\sum_{f\in\mathcal{F}} P_{YF}(Y=y, F=f)T(f,b)$$

$$= \sum_{y\in\mathcal{Y}}\sum_{f\in\mathcal{F}} P_{YF}(Y=y, F=f)\left(\sum_{b\in\mathcal{B}} T(f,b)\right)$$

$$= \sum_{y\in\mathcal{Y}}\sum_{f\in\mathcal{F}} P_{YF}(Y=y, F=f) = 1. \tag{13}$$

Also, we can show that $Q^* \in \Delta_P$ since:

$$Q^*_{YB}(Y=y, B=b) = \sum_{f\in\mathcal{F}} P_{YF}(Y=y, F=f)T(f,b) = P_{YB}(Y=y, B=b), \tag{14}$$

which holds since such a row-stochastic matrix $T$ exists. Also, we have:

$$Q^*_{YF}(Y=y, F=f) = \sum_{b\in\mathcal{B}} P_{YF}(Y=y, F=f)T(f,b) = P_{YF}(Y=y, F=f), \tag{15}$$

which holds since $T$ is row-stochastic.

Then, $\mathrm{Uni}(Y{:}B|F) = \min_{Q\in\Delta_P} \mathrm{I}_Q(Y;B|F) \leq \mathrm{I}_{Q^*}(Y;B|F) = 0$.

$\square$

## C.2 Proof of Theorem 1

For the first claim, notice that $\mathrm{Uni}(Y{:}B|F) = \mathrm{I}(Y;B) - \mathrm{Red}(Y{:}B, F)$ (from equation 4) and $\mathrm{Red}(Y{:}B, F) \geq 0$ (non-negativity of PID, see Lemma 4). Thus,

$$\mathrm{Uni}(Y{:}B|F) \leq \mathrm{I}(Y;B).$$

For the second claim, we will use Lemma 7. $\mathrm{Uni}(Y{:}B|F) = 0$ if and only if there exists a row-stochastic matrix $T \in [0,1]^{|\mathcal{F}|\times|\mathcal{B}|}$ such that: $P_{YB}(Y=y, B=b) = \sum_{f\in\mathcal{F}} P_{YF}(Y=y, F=f)T(f,b)$ for all $y \in \mathcal{Y}$ and $b \in \mathcal{B}$. The existence of such a row-stochastic matrix is equivalent to Blackwell Sufficiency as per Definition 2 from (Blackwell, 1953).

For the third claim, first observe that if $B' = B \cup W$, then $B$ can be written as a local operation on $B'$, i.e., $B = f(B')$. Thus, from Lemma 5, we have:

$$\mathrm{Uni}(Y{:}B|F) \leq \mathrm{Uni}(Y{:}B'|F). \tag{16}$$

Next, observe that since $F' = F\backslash W$, then from Lemma 6, we have:

$$\mathrm{Uni}(Y{:}B'|F) = \mathrm{Uni}(Y{:}B'|F', W) \leq \mathrm{Uni}(Y{:}B'|F'). \tag{17}$$

Combining equation 16 and equation 17, we have the claim

$$\mathrm{Uni}(Y{:}B|F) \leq \mathrm{Uni}(Y{:}B'|F').$$

## C.3 Proof of Additional Results

### C.3.1 Proof of Lemma 1

*Proof of Lemma 1.* Here, $B = Y + N$ and $F = Y + N$ where $Y$ and $N$ are independent. Any optimal predictor is a function of the inputs $F$ and $B$, i.e., $\hat{Y} = f(F, B)$. Since $F = B$, this function can always be rewritten as a function of $B$ alone or $F$ alone.

Next, we will show that only the redundant information $\text{Red}(Y{:}B, F)$ is positive and all other PID terms $\text{Uni}(Y{:}B|F)$, $\text{Uni}(Y{:}F|B)$, and $\text{Syn}(Y{:}F, B)$ are zero.

Here $\text{I}(Y; B|F) = \text{I}(Y; F|B) = 0$ since $B = F$.

$$\text{I}(Y; B|F) = H(B|F) - H(B|Y, F) = 0.$$

According to the Definition 1 and non-negativity of PID terms, $\text{Uni}(Y{:}B|F) = \text{I}(Y; B|F) - \text{Syn}(Y{:}F, B) \leq \text{I}(Y; B|F) = 0$.

Similarly, we have, $\text{Uni}(Y{:}F|B) \leq \text{I}(Y; F|B) = 0$.

Then, $\text{Syn}(Y{:}F, B) = \text{I}(Y; F|B) - \text{Uni}(Y{:}F|B)$ (from equation 5) is also 0.

Now, $\text{Red}(Y{:}B, F) = \text{I}(Y; B) - \text{Uni}(Y{:}B|F) = \text{I}(Y; B) = H(Y) - H(Y|B)$ which is positive as long as there is a significant dependence between $Y$ and $B$. $\qquad\square$

### C.3.2 Proof of Lemma 2

We first include another lemma that will be useful in proving our main result.

**Lemma 8** (Noisy Feature). *Let $A = Y + N$ where $Y \sim Bern(1/2)$ is a random variable taking values $+1$ or $-1$ and the noise $N \sim \mathcal{N}(0, \sigma_N^2)$ is a Gaussian random variable independent of $Y$. Then, the mutual information*

$$\text{I}(Y; A) \leq \frac{1}{2}\log_2\left(1 + \frac{1}{\sigma_N^2}\right).$$

*Proof.*

$$\text{I}(Y; A) = H(A) - H(A|Y) = H(Y + N) - H(Y + N|Y) \tag{18}$$
$$= H(Y + N) - H(N|Y) \tag{19}$$
$$= H(Y + N) - H(N), \text{ since } N \perp\!\!\!\perp Y \tag{20}$$
$$\overset{(a)}{\leq} \frac{1}{2}\log_2 2\pi e \left(1 + \sigma_N^2\right) - \frac{1}{2}\log_2 2\pi e \left(\sigma_N^2\right) \tag{21}$$
$$= \frac{1}{2}\log_2\left(1 + \frac{1}{\sigma_N^2}\right). \tag{22}$$

Here (a) holds because the entropy of $Y + N$ is bounded by $\frac{1}{2}\log_2 2\pi e \left(1 + \sigma_N^2\right)$ (proved in Cover & Thomas (2012, Theorem 8.6.5)). We also refer to Cover & Thomas (2012, Chapter 9) for a discussion on Gaussian channels. $\qquad\square$

If we keep the distribution of $Y$ fixed and vary the noise variance $\sigma_N^2$, then we will observe a decreasing trend of $\text{I}(Y; B)$ with increasing $\sigma_N^2$. Fig.15 shows the exact trend where $Y$ is a Bernoulli random variable.

*Proof of Lemma 2.* Here $B = N$ and $F = Y + N$ where $Y \sim Bern(1/2)$ takes values $+1$ or $-1$, and the noise $N \sim \mathcal{N}(0, \sigma_N^2)$ with $N \perp\!\!\!\perp Y$ and $\sigma_N^2 \gg 1$.

First observe that the predictor $\hat{Y} = f(B, F) = F - B = Y$. Thus, it is perfectly predictive of $Y$, and is an optimal predictor.

Now, we will compute the values of the PID terms and show that $\text{Syn}(Y{:}B, F) > 0$ and all the other three PID terms are negligible.

Since $B \perp\!\!\!\perp Y$, we have $\text{I}(Y; B) = 0$.

Since $F = Y + N$, we use Lemma 8 to first show that: $\text{I}(Y; F) \leq \frac{1}{2}\log_2\left(1 + \frac{1}{\sigma_N^2}\right)$. Now, as the variance $\sigma_N^2$ becomes high, we have $\text{I}(Y; F) \approx 0$.

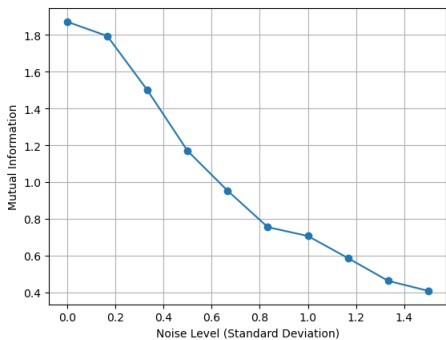

Figure 15: Mutual Information vs. Noise Level ($Y$ is Bernoulli)

Since $N \perp\!\!\!\perp Y$, we have $\mathrm{I}(Y; B) = 0$. Now, from equation 4, we have

$$\mathrm{I}(Y; B) = \mathrm{Uni}(Y{:}B|F) + \mathrm{Red}(Y{:}B, F) = 0. \tag{23}$$

According to Lemma 4, $\mathrm{Uni}(Y{:}B|F)$ and $\mathrm{Red}(Y{:}B, F)$ are nonnegative. As their summation is 0, each term should be 0 as well, i.e., $\mathrm{Uni}(Y{:}B|F) = 0$ and $\mathrm{Red}(Y{:}B, F) = 0$.

Again, since $N$ has a high variance, we have (from Lemma 8):

$$\mathrm{I}(Y; F) \leq \frac{1}{2} \log_2 \left(1 + \frac{1}{\sigma_N^2}\right) \approx 0. \tag{24}$$

This leads to $\mathrm{Uni}(Y{:}F|B) = \mathrm{I}(Y; F) - \mathrm{Red}(Y{:}B, F) \leq \frac{1}{2} \log_2 \left(1 + \frac{1}{\sigma_N^2}\right) \approx 0$.

However, $\mathrm{I}(Y; F|B) = H(Y|B) - H(Y|B, F) = H(Y|N) - H(Y|Y + N, N) = H(Y)$ which is positive and significant. This holds because $H(Y|Y + N, N) = 0$ since $Y$ is completely determined by $Y + N$ and $N$ together.

Now,

$$\mathrm{Syn}(Y{:}B, F) = \mathrm{I}(Y; F|B) - \mathrm{Uni}(Y{:}F|B) \geq H(Y) - \frac{1}{2} \log_2 \left(1 + \frac{1}{\sigma_N^2}\right) \approx H(Y). \tag{25}$$

$\square$

### C.3.3 Proof of Lemma 3

*Proof of Lemma 3.* Here, the input feature $X = (F, B)$. Observe that we have the following conditional distributions: $X|_{Y=0} \sim \mathcal{N}([0\ 0], \begin{bmatrix} \sigma_{N_F}^2 & 0 \\ 0 & \sigma_{N_B}^2 \end{bmatrix})$, and $X|_{Y=1} \sim \mathcal{N}([1\ 1], \begin{bmatrix} \sigma_{N_F}^2 & 0 \\ 0 & \sigma_{N_B}^2 \end{bmatrix})$. For simplicity, assume $P(Y = 0) = P(Y = 1)$. We let $\Sigma = \begin{bmatrix} \sigma_{N_F}^2 & 0 \\ 0 & \sigma_{N_B}^2 \end{bmatrix}$.

For the Bayes optimal classifier at the decision boundary, we have:

$$
\begin{aligned}
&P(X|Y=0) = P(X|Y=1)\\
&\Rightarrow \log(P(X|Y=0)) = \log(P(X|Y=1))\\
&\Rightarrow -\frac{1}{2}X\Sigma^{-1}X^{\top} = -\frac{1}{2}(X - [1\ 1])\Sigma^{-1}(X - [1\ 1])^{\top}\\
&\Rightarrow \frac{\|F\|_2^2}{\sigma_{N_F}^2} + \frac{\|B\|_2^2}{\sigma_{N_B}^2} = \frac{\|F-1\|_2^2}{\sigma_{N_F}^2} + \frac{\|B-1\|_2^2}{\sigma_{N_B}^2}\\
&\Rightarrow \frac{F}{\sigma_{N_F}^2} + \frac{B}{\sigma_{N_B}^2} = \frac{1}{2\sigma_{N_F}^2} + \frac{1}{2\sigma_{N_B}^2}
\end{aligned}
$$

This is the decision boundary for the Bayes optimal classifier. Thus, we can show that when $\sigma_{N_B}^2 \gg \sigma_{N_F}^2$, the boundary relies heavily on core feature $F$. Similarly, when $\sigma_{N_F}^2 \gg \sigma_{N_B}^2$, the boundary relies heavily on spurious feature $B$. Also refer to Fig. 3 (first two cases) for a pictorial illustration of how the optimal classifier behaves.

Next, observe that when $\sigma_{N_F}^2 \gg \sigma_{N_B}^2$, we have $\mathrm{I}(Y;B) > \mathrm{I}(Y;F)$ with strict equality (see Lemma 8).

From the definition of PID, $\mathrm{I}(Y;B) = \mathrm{Uni}(Y{:}B|F) + \mathrm{Red}(Y{:}B,F)$ and $\mathrm{I}(Y;F) = \mathrm{Uni}(Y{:}F|B) + \mathrm{Red}(Y{:}B,F)$.

Since $\mathrm{I}(Y;B) > \mathrm{I}(Y;F)$, we therefore have:

$$
\mathrm{Uni}(Y{:}B|F) + \mathrm{Red}(Y{:}B,F) > \mathrm{Uni}(Y{:}F|B) + \mathrm{Red}(Y{:}B,F).
$$

This leads to $\mathrm{Uni}(Y{:}B|F) > \mathrm{Uni}(Y{:}F|B) \geq 0$ since each PID term is nonnegative.

□

### C.4 Additional Details on Dimensionality Reduction

The representation loss is the mean square error between the input of the encoder $x$ and the output of the decoder $x'$ defined as $L_r = \|x - x'\|_2^2$. The cluster centers $\{\mu_j\}_1^K$ (trainable weights of clustering layer) and embedded point $z_i$ (output of the encoder) are used to calculate the soft label $q_{ij} = \frac{(1+\|z_i - \mu_j\|^2)^{-1}}{\sum_j (1+\|z_i - \mu_j\|^2)^{-1}}$ where $q_{ij}$ is the $j$th entry of the soft label $q_i$, denoting the probability of $z_i$ belonging to cluster $\mu_j$. The clustering loss $L_c$ is the KL divergence between the soft assignments ($q_i$) and an auxiliary distribution ($p_i$). First, the autoencoder is pre-trained using only $L_r$ to initialize the auxiliary distribution, and the cluster centers are initialized by performing k-means on the embeddings of all images. After pretaining, the cluster centers and autoencoder weights are updated with the joint loss $L$ iteratively while the auxiliary distribution is only updated after $T$ iterations.

## D Appendix to Experiments

This section includes additional results and figures for a more comprehensive understanding.

### D.1 Additional Details on Automatic Segmentation of Features

Segmentation, a component of our Spurious Disentangler, plays a pivotal role in identifying core features from spurious ones. Identifying spurious features (pixels) without any additional information is challenging in image datasets, particularly if they lack group labels. However, in supervised classification tasks, the availability of target labels corresponding directly to the goal of the classification task (and hence some partial knowledge of what the core features should be, if not the exact pixels) often offers a practical workaround. Specifically, one can leverage automatic segmentation to at least perform object detection and

choose the most relevant objects as the "core". Then, the regions of an image not associated with the "core" objects can often be considered a subset of spurious features.

Advances in Open-Vocabulary Semantic Segmentation (OVSS) have significantly reduced the dependence on task-specific training by enabling generalization to unseen categories without requiring labeled data. To leverage these advancements, we employ CLIPSeg (Lüddecke & Ecker, 2022), a state-of-the-art OVSS model, to generate masks for various objects in a zero-shot manner using partial knowledge of the classification task in mind. For instance, in the Waterbird dataset, we specify the prompt "bird" to obtain a mask for the bird object. This approach utilizes publicly available fine-grained weights, enabling efficient and accurate segmentation without additional labeled data. The generated mask is applied to the original image to extract the foreground, while the background is obtained by multiplying the original image by $1 - mask$, as illustrated in Fig. 12.

Thus, our proposed technique of dataset evaluation can be applied in conjunction with such automatic segmentation methods to any image dataset where the group information is not available, enabling us to first identify an approximation of the core features using partial knowledge of the target objects for the classification task, and then explain the nature of spurious patterns.

## D.2 Additional Results

Our explainability framework is preemptive or anticipative of spuriousness using just the dataset before training the model. The goal of our experiments is to show broad agreement between our anticipations from the dataset before training any model and the post-training behavior of actual models (when trained regularly to optimize performance without doing anything else specifically targeted towards avoiding spurious features). Apart from Worst-Group Accuracy, we also observe the Grad-CAM visualizations to check if the model demonstrates a stronger emphasis on the relevant core features or not (see Fig. 10, 21, 18). To further justify this, we calculate the intersection-over-union (IoU) metric (Rezatofighi et al., 2019) over the entire test Waterbird dataset. Table 1 shows that when the dataset is modified from unbalanced to the other variants, the IoU score increases. The IoU score is calculated using the ground-truth segmentation masks of birds and the masks obtained from the Grad-CAM explanation.

Table 1: Intersection over Union (IoU) between ground truth and Grad-CAM masks on the WATERBIRD dataset

| Test Group | Unbalanced | Class Balanced | Group Balanced | Addition | Concatenation |
|---|---|---|---|---|---|
| Minority Group | 0.22 | 0.29 | 0.24 | 0.28 | 0.32 |
| All Groups | 0.19 | 0.23 | 0.22 | 0.29 | 0.30 |

Table 2 shows a comparison between our proposed measure of spuriousness $M_{sp}$ and other possible measures.

## D.3 Additional Details on Clustering

At the dimensionality reduction stage, we must choose an appropriate number of clusters. We compute PID values for cluster counts of 5, 10, and 20. As shown in Table 3, the PID components —and consequently the spuriousness measure $M_{sp}$— are sensitive to the number of clusters. However, when moving from the unbalanced to the class-balanced setting, $M_{sp}$ consistently decreases across all cluster counts, indicating that essential information is retained despite dimensionality reduction. A similar sensitivity analysis on the CMNIST dataset (see Fig. 16) confirms this pattern. We observe that for CMNIST with two target classes, the computational time increases significantly — from approximately 6.8 seconds (cluster size 5) to 470.63 seconds (cluster size 50) — when the flip rate is set to 0% (see Table 4). However, most importantly, we observe that for more than 10 clusters, the convex optimization for PID calculation sometimes fails to converge even after 50,000 iterations. Based on these results and previous work (Guo et al., 2017), we choose 10 clusters as a trade-off between preserving sufficient information and ensuring computational efficiency and faster convergence.

Table 2: Comparison of the proposed spuriousness measure ($M_{sp}$) with alternative metrics across dataset variants

| Dataset | Measure | Unbalanced | Class Balanced | Group Balanced | Addition | Concatenation |
|---|---|---|---|---|---|---|
| WATERBIRD | $I(Y;B)$ | 0.1726 | 0.0315 | 0.0028 | 0.0005 | 0.0002 |
| | $I(Y;B) - I(Y;F)$ | 0.1669 | 0.0298 | -0.0089 | -0.0052 | -0.0054 |
| | Proposed $M_{sp}$ | 0.1486 | 0.0185 | -0.0322 | -0.0208 | -0.0195 |
| DOMINOES 1.0 | $I(Y;B)$ | 0.1882 | — | 0.0005 | 0.0010 | 0.0010 |
| | $I(Y;B) - I(Y;F)$ | 0.1728 | — | -0.0010 | -0.0203 | -0.0144 |
| | Proposed $M_{sp}$ | 0.1660 | — | -0.0165 | -0.0279 | -0.0207 |
| DOMINOES 2.0 | $I(Y;B)$ | 0.5913 | — | 0.2610 | 0.0002 | 0.0001 |
| | $I(Y;B) - I(Y;F)$ | 0.5619 | — | 0.2462 | -0.0426 | -0.0477 |
| | Proposed $M_{sp}$ | 0.5557 | — | 0.2237 | -0.0501 | -0.0574 |
| CELEBA | $I(Y;B)$ | 0.0238 | 0.0005 | 0.0151 | — | — |
| | $I(Y;B) - I(Y;F)$ | -0.3038 | -0.3713 | -0.4051 | — | — |
| | Proposed $M_{sp}$ | -0.3091 | -0.3775 | -0.4797 | — | — |
| SPAWRIOUS | $I(Y;B)$ | 0.0437 | — | 0.0096 | — | — |
| | $I(Y;B) - I(Y;F)$ | 0.0012 | — | -0.0056 | — | — |
| | Proposed $M_{sp}$ | -0.0007 | — | -0.0176 | — | — |

Table 3: PID components for the Waterbird dataset across varying cluster sizes under unbalanced and class-balanced conditions.

| Unbalanced | | | | | |
|---|---|---|---|---|---|
| # Clusters | $Red(Y{:}F, B)$ | $Uni(Y{:}B \mid F)$ | $Uni(Y{:}F \mid B)$ | $Syn(Y{:}F, B)$ | $M_{sp}$ |
| 5 | 0.0065 | 0.1220 | 0.0000 | 0.0085 | 0.1135 |
| 10 | 0.0057 | 0.1669 | 0.0000 | 0.0184 | 0.1485 |
| 20 | 0.0025 | 0.1736 | 0.0000 | 0.0163 | 0.1573 |
| Class Balanced | | | | | |
| # Clusters | $Red(Y{:}F, B)$ | $Uni(Y{:}B \mid F)$ | $Uni(Y{:}F \mid B)$ | $Syn(Y{:}F, B)$ | $M_{sp}$ |
| 5 | 0.0008 | 0.0221 | 0.0000 | 0.0012 | 0.0209 |
| 10 | 0.0016 | 0.0300 | 0.0001 | 0.0114 | 0.0185 |
| 20 | 0.0008 | 0.0128 | 0.0000 | 0.0097 | 0.0031 |

Table 4: PID estimation time with varying number of clusters for CMNIST (flip rate 0%).

| Number of Clusters | Time (seconds) |
|---|---|
| 5 | 6.80 |
| 10 | 18.69 |
| 20 | 65.12 |
| 30 | 153.29 |
| 50 | 470.63 |

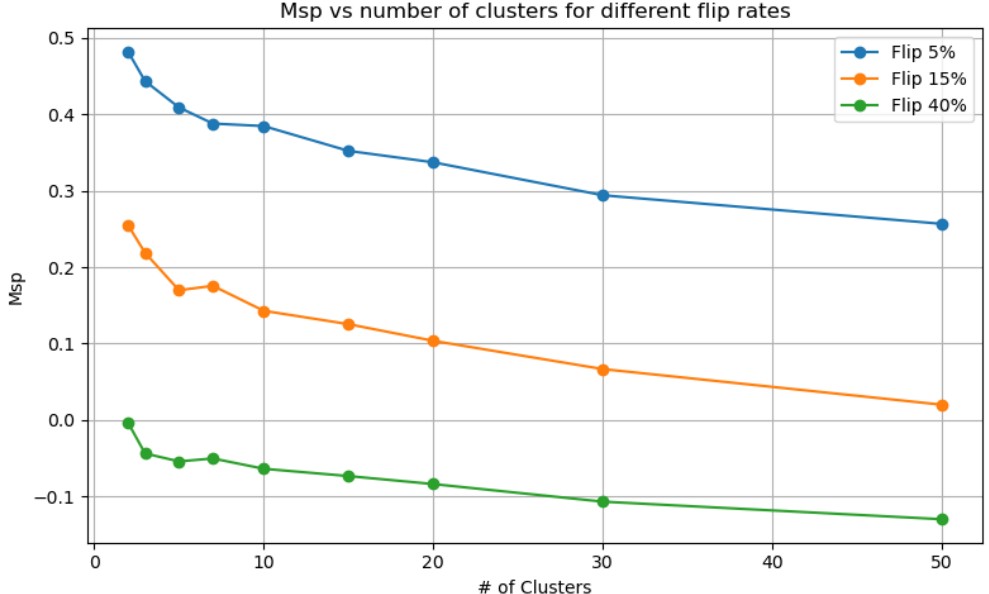

Figure 16: $M_{sp}$ versus number of clusters for different label flip percentages. As the number of clusters increases, $M_{sp}$ tends to decrease, and higher flip rates result in lower $M_{sp}$ values, indicating decreased spuriousness.

## D.4 Additional Details on Datasets

Table 5: Summary of the datasets

| Dataset | Split | Group 00 | Group 01 | Group 10 | Group 11 |
|---|---|---|---|---|---|
| WATERBIRD | Train | 3,498 | 184 | 56 | 1,057 |
| | Validation | 467 | 466 | 133 | 133 |
| | Test | 2,255 | 2,255 | 642 | 642 |
| DOMINOES 1.0 | Train | 3,750 | 1,250 | 1,250 | 3,750 |
| | Test | 473 | 507 | 507 | 473 |
| DOMINOES 2.0 | Train | 3,000 | 500 | 1,250 | 300 |
| | Test | 245 | 490 | 245 | 490 |
| ADULT | Train | 10,116 | 15,930 | 1,214 | 6,929 |
| | Test | 4,307 | 6,802 | 555 | 2,989 |
| CELEBA | Train | 11,111 | 8,305 | 4,003 | 188 |
| | Test | 1,391 | 997 | 525 | 18 |
| SPAWRIOUS | Train | 3,072 | 2,275 | 175 | 1,056 |
| | Test | 96 | 893 | 2,993 | 2,112 |

Table 6: PID values across dataset variants

| Dataset | Variation | Red($Y$:$F, B$) | Uni($Y$:$B\|F$) | Uni($Y$:$F\|B$) | Syn($Y$:$F, B$) | $M_{sp}$ |
|---|---|---|---|---|---|---|
| WATERBIRD | Unbalanced | 0.0057 | 0.1669 | 0.0000 | 0.0184 | 0.1486 |
| | Class Balanced | 0.0016 | 0.0300 | 0.0001 | 0.0114 | 0.0185 |
| | Group Balanced | 0.0026 | 0.0001 | 0.0091 | 0.0233 | -0.0322 |
| | Addition | 0.0004 | 0.0001 | 0.0053 | 0.0156 | -0.0208 |
| | Concatenation | 0.0002 | 0.0001 | 0.0055 | 0.0140 | -0.0195 |
| ADULT | Unbalanced | 0.0374 | 0.0000 | 0.0661 | 0.0267 | -0.0928 |
| | Group Balanced | 0 | 0 | 0.1163 | 0.0090 | -0.1252 |
| DOMINOES 1.0 | Unbalanced | 0.0154 | 0.1728 | 0.0000 | 0.0068 | 0.1660 |
| | Group Balanced | 0.0003 | 0.0002 | 0.0013 | 0.0155 | -0.0165 |
| | Addition | 0.0009 | 0.0000 | 0.0203 | 0.0076 | -0.0279 |
| | Concatenation | 0.0009 | 0.0000 | 0.0144 | 0.0063 | -0.0207 |
| DOMINOES 2.0 | Unbalanced | 0.0294 | 0.5619 | 0.0000 | 0.0061 | 0.5557 |
| | Group Balanced | 0.0148 | 0.2462 | 0.0000 | 0.0225 | 0.2237 |
| | Addition | 0.0001 | 0.0000 | 0.0426 | 0.0075 | -0.0501 |
| | Concatenation | 0.0001 | 0.0000 | 0.0477 | 0.0096 | -0.0574 |
| CELEBA | Unbalanced | 0.0238 | 0.0000 | 0.3038 | 0.0053 | -0.3091 |
| | Class Balanced | 0.0000 | 0.0000 | 0.3713 | 0.0063 | -0.3775 |
| | Group Balanced | 0.0151 | 0.0000 | 0.4051 | 0.0746 | -0.4797 |
| SPAWRIOUS | Unbalanced | 0.0396 | 0.0041 | 0.0029 | 0.0019 | -0.0007 |
| | Group Balanced | 0.0089 | 0.0007 | 0.0063 | 0.0121 | -0.0176 |

### D.4.1 Both core and spurious features available

WATERBIRD: The Waterbird dataset (Wah et al., 2011) is a popular spurious correlation benchmark. The task is to classify the type of bird (waterbird = 1, landbird = 0). However, there exists a spurious correlation between the backgrounds (water = 1, land = 0) and the labels (bird type). The two types of backgrounds and foregrounds result in a total of four groups. A summary of the Waterbird dataset is given in Table 5, and see Fig. 17 for examples of the dataset. We additionally consider four random training splits with group counts [2096 212 102 2385], [3493 213 58 1031], [2632 420 150 1593], and [1363 942 1469 1021] to observe the correlation between the measure of spuriousness $M_{sp}$ and W.G. Acc. (%) (see Fig.7) in diverse settings. We use Spurious Disentangler to calculate the PID values. The hyperparameters are as follows: a batch size of 64, a learning rate of 0.001, a CosineAnnealingLR scheduler, an Adam optimizer with a weight decay of 0.0001, 50 pretraining epochs, followed by 100 epochs of additional training. When fine-tuning ResNet-50 we use the following hyperparameters: batch size of 64, learning rate of 0.0001, CosineAnnealingLR scheduler, stochastic gradient descent (SGD) optimizer with a weight decay of 0.0001, binary cross-entropy as the loss function, and 100 epochs. See Table 6 for the details of the PID values, and Table 7 for the worst-group accuracies of different variants of the Waterbird dataset.

DOMINOES: Dominoes is a synthetic dataset created by combining handwritten digits (zero and one) from MNIST (Deng, 2012) and images of cars and trucks from CIFAR10 (Krizhevsky et al., 2009) (digit 0 or 1 at the top, car (= 0) or truck (= 1) at the bottom of an image). We make two versions of this synthetic dataset, namely Dominoes 1.0 and Dominoes 2.0, inducing different degrees of sampling biases. The task is to classify whether the image contains a car or a truck; hence, the car or truck corresponds to the core features (foreground). On the other hand, the digits are considered as the spurious features (background).

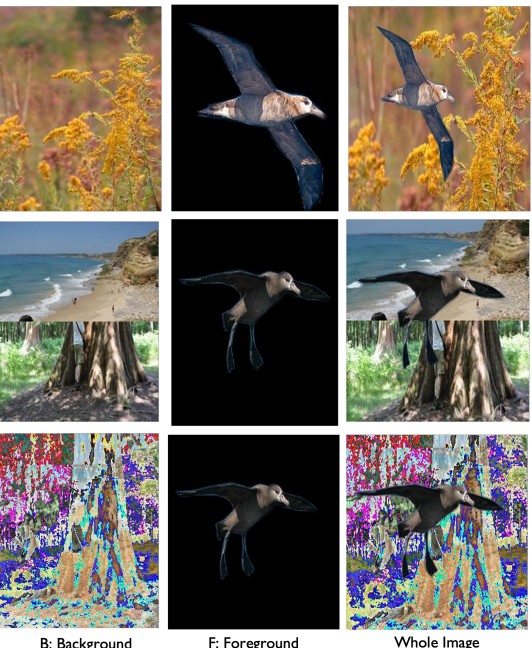

B: Background    F: Foreground    Whole Image

Figure 17: Samples of Waterbird dataset (original, concatenation, and addition).

Table 7: Worst-group accuracy (%) across dataset variants

| Dataset | Unbalanced | Class Balanced | Group Balanced | Addition | Concatenation |
|---|---|---|---|---|---|
| WATERBIRD | $45.92 \pm 1.43$ | $74.49 \pm 0.58$ | $85.82 \pm 0.71$ | $88.18 \pm 2.17$ | $92.60 \pm 0.39$ |
| DOMINOES 1.0 | $86.29 \pm 4.44$ | — | $90.19 \pm 1.23$ | $94.42 \pm 0.24$ | $96.06 \pm 0.39$ |
| DOMINOES 2.0 | $78.78 \pm 1.02$ | — | $88.06 \pm 1.12$ | $86.74 \pm 1.22$ | $90.72 \pm 3.37$ |
| CELEBA | $71.41 \pm 0.81$ | $85.29 \pm 2.94$ | $98.34 \pm 1.66$ | — | — |
| SPAWRIOUS | $91.91 \pm 1.94$ | — | $95.24 \pm 0.28$ | — | — |

The summary of Dominoes 1.0 and Dominoes 2.0 is given in Table 5. Fig. 19 shows the examples of original, addition, and concatenation variants of the dataset. For PID calculation, the hyperparameters are as follows: a batch size of 8, a learning rate of 0.001, a CosineAnnealingLR scheduler, an Adam optimizer with a weight decay of 0.0001, 100 pretraining epochs, followed by 50 epochs of additional training. See Table 6 for the details of PID values and $M_{sp}$.

TABULAR DATASET: ADULT: The applicability of our proposed framework goes beyond images and can also be applied for explainability on tabular datasets. We perform an experiment on the Adult (Becker & Kohavi, 1996) dataset. The task is to predict whether the annual income of an individual exceeds \$50k per year or not ($> 50k = 1, <= 50k = 0$). Here we consider "gender" as a spurious feature vector (male $= 1$, female $= 0$) and "age", "education-num", "hours-per-week" jointly as a core feature matrix. After performing k-means clustering, we use the estimation module (DIT package) to calculate PID values with core features, spurious features, and the target label. Table 6 shows the values for redundancy, unique information, and synergy.

We train the XGBoost (Chen & Guestrin, 2016) model for the prediction task and calculate the worst-group accuracy which corresponds to the accuracy of the minority group (see Table 5, minority group 10 corresponds to female individuals with >50k income.).

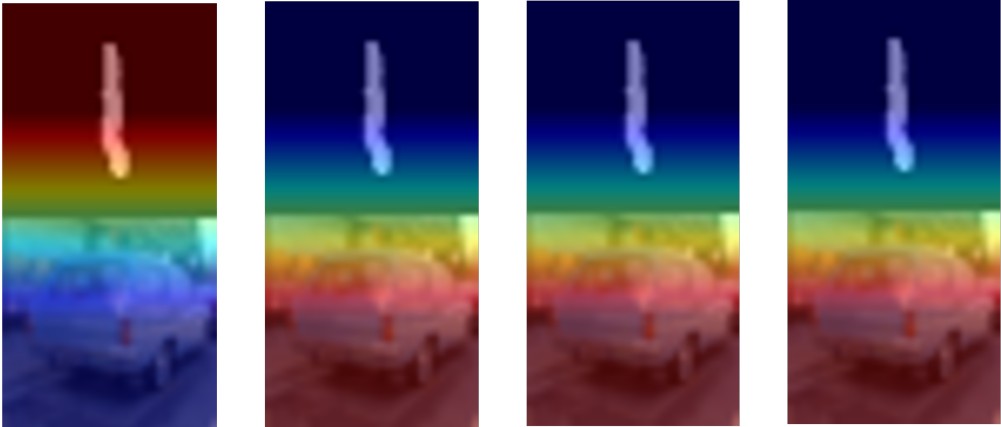

Figure 18: Examples of Grad-CAM images Dominoes dataset: Observe that for the unbalanced dataset (1st from left), the model adds more emphasis (red regions) to the digits (background) while in the group-balanced, addition and concatenation versions (2nd, 3rd, and 4th from left), the car (foreground) is more emphasized.

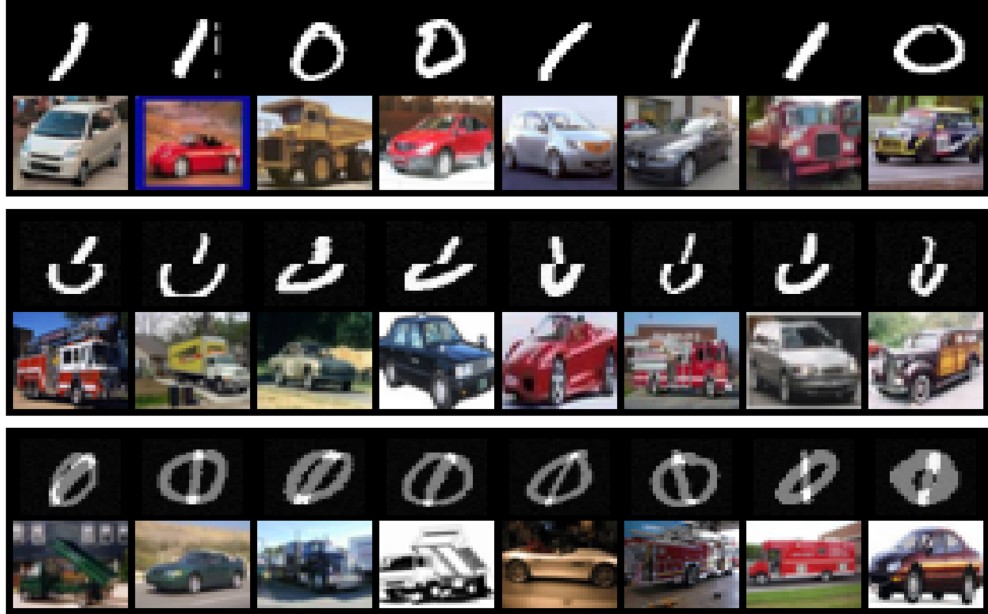

Figure 19: Samples of Dominoes dataset (original, concatenation, and addition).

### D.4.2 Only core features available

CELEBA: CelebA is another popular dataset for spurious correlation benchmarking, which consists of images of male-female celebrities. We use a subset of this dataset, namely CelebAMask-HQ (Lee et al., 2020), to utilize the segmentation mask of the hair while calculating the PID values. The objective is to identify blonde ($= 1$) and non-blonde ($= 0$) hair. However, there exists a spurious correlation between gender (men ($= 1$), women ($= 0$)) and the label, which makes the model focus on the face rather than the hair to classify the color of the hair (Moayeri et al., 2023). We consider hair as the foreground and anything but hair as the background. We do not perform background mixing for this dataset since it is not practical to add or concatenate two faces randomly. The summary of the CelebA (Lee et al., 2020) dataset is given in Table 5. The steps and hyperparameters for calculating PIDs are the same as in the Waterbird dataset. See Fig. 20 for

the examples of dataset samples. The details of PID values and worst-group accuracies for several variations of this dataset are shown in Table 6 and Table 7, respectively.

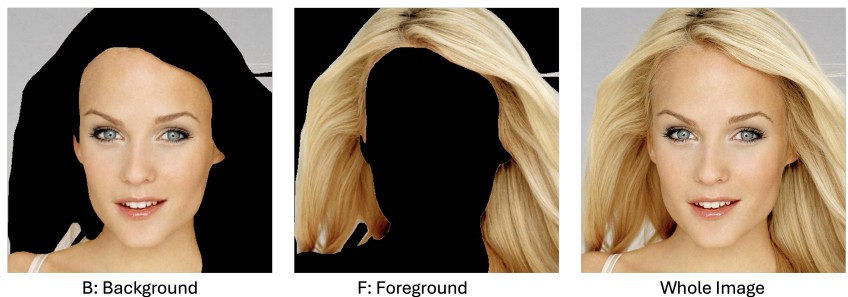

Figure 20: Samples of the CelebA dataset.

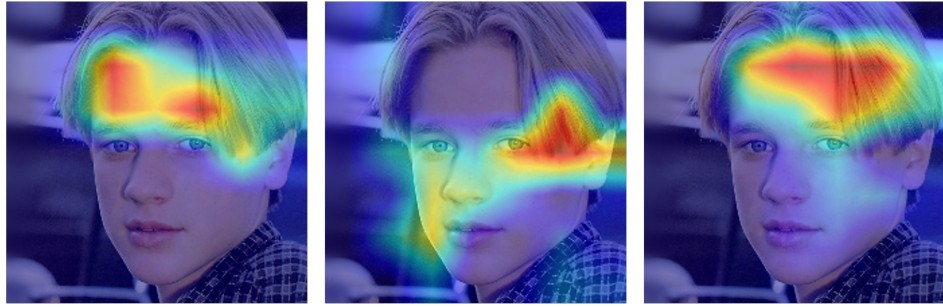

Figure 21: Examples of Grad-CAM images CelebA dataset: Observe that for the unbalanced dataset (1st from left), the model adds more emphasis (red regions) to the face (background) while in the class-balanced and group-balanced (2nd and 3rd), the hair (foreground) is more emphasized.

### D.4.3 Segmentation to obtain features

SPAWRIOUS: Spawrious (Lynch et al., 2023) is a synthetic image dataset created by employing a text-to-image model. We use a subset of this dataset where we classify dog breeds - dachshund ($= 0$) and labrador ($= 1$). We select the subset so that most of the dachshunds are on beach ($= 0$) backgrounds and the rest are on desert ($= 1$) backgrounds. The summary of the subset of the Spawrious dataset (Lynch et al., 2023) that we use for our experiment is given in Table 5. The samples of this dataset are shown in Fig. 22. We use a segmentation model, namely Feature Pyramid Network (FPN) (Lin et al., 2017) with ResNet-34 (He et al., 2016) encoder pre-trained with Oxford-IIIT Pet Dataset to create the segmentation mask of the dogs of our dataset. Using this mask, we separate the foreground "dog" from the background. After having background and foreground, we use principal component analysis (PCA) (Maćkiewicz & Ratajczak, 1993) followed by k-means clustering to have a discrete lower-dimensional representation. We do not use our autoencoder module since, for this dataset, a simpler dimensionality reduction also seems to have a low reconstruction loss. Then we use our estimation module to calculate the PID values and $M_{sp}$. Tables 6 and Table 7 show all PID values along with the measure and the worst-group accuracy, respectively.

### D.4.4 Non-spatial spuriousness

COLORED MNIST: Colored MNIST (CMNIST) is a synthetic dataset derived from MNIST. Whereas MNIST images are grayscale, each image in CMNIST is colored either red or green in a way that correlates spuriously with the class label. We define two environments (one training, one test) from MNIST, transforming each example as follows: first, assign a preliminary binary label y to the image based on the digit: $y = 0$ for digits $0 - 4$ and $y = 1$ for $5 - 9$. Second, sample the color id $z$ by flipping $y$ with probability $p_e$ (flip probability). In the test one, the flip probability is kept 0.9. Finally, color the image red if $z = 1$ or green if $z = 0$ (see

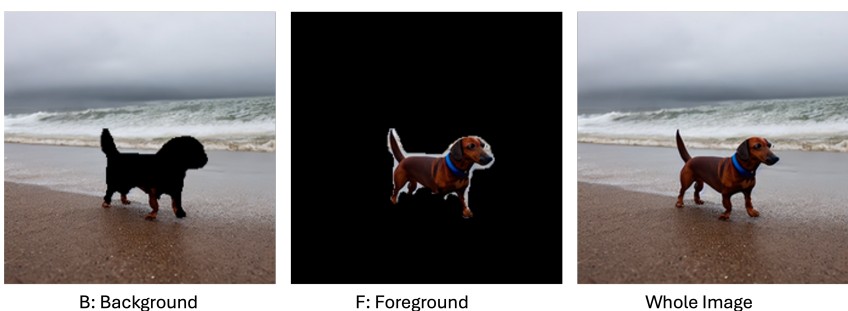

Figure 22: Samples of the subset of the Spawrious dataset we use in this work.

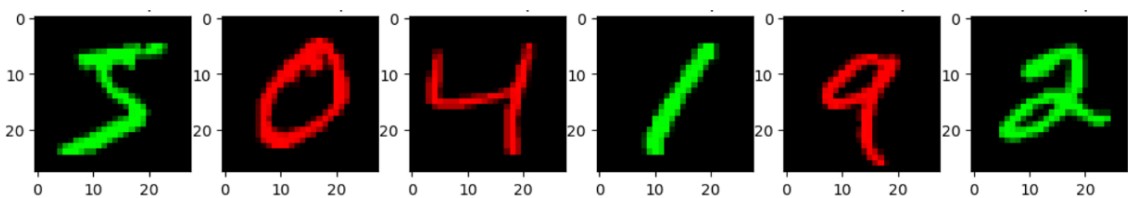

Figure 23: Samples of the CMNIST dataset used in this work.

Table 8: Comparison of spuriousness measure $M_{sp}$ and worst-group accuracy (%) for multiclass and binary classification on CMNIST under different flip probabilities.

| Flip (%) | Binary | | Multiclass | |
|---|---|---|---|---|
| | $M_{sp}$ | W.G. Acc. (%) | $M_{sp}$ | W.G. Acc. (%) |
| 0.0 | 0.58581 | 0.13±0.15 | 0.40220 | 0.19±0.14 |
| 2.5 | 0.45287 | 78.77±1.18 | 0.27035 | 83.88±1.66 |
| 5.0 | 0.38470 | 87.96±0.37 | 0.17788 | 86.96±2.97 |
| 7.5 | 0.30933 | 92.94±0.54 | 0.07443 | 93.11±0.84 |
| 10.0 | 0.24979 | 94.97±0.85 | 0.03366 | 93.33±0.97 |
| 12.5 | 0.19956 | 95.00±0.43 | -0.00945 | 94.46±0.80 |
| 15.0 | 0.14296 | 95.73±0.46 | -0.06050 | 94.98±0.37 |
| 17.5 | 0.10727 | 96.13±0.37 | -0.11422 | 94.83±0.43 |
| 20.0 | 0.07481 | 96.49±0.58 | -0.14608 | 95.66±0.39 |
| 22.5 | 0.04553 | 97.21±0.28 | -0.19066 | 95.99±0.68 |
| 25.0 | 0.02400 | 97.62±0.35 | -0.20659 | 96.24±0.05 |
| 27.5 | -0.01175 | 97.73±0.38 | -0.22717 | 95.87±0.20 |
| 30.0 | -0.01043 | 97.78±0.29 | -0.25008 | 96.05±0.52 |
| 32.5 | -0.03903 | 98.04±0.25 | -0.27048 | 96.55±0.31 |
| 35.0 | -0.05213 | 98.17±0.22 | -0.28146 | 96.86±0.53 |
| 37.5 | -0.06233 | 98.34±0.36 | -0.28625 | 96.11±0.22 |
| 40.0 | -0.06381 | 98.42±0.26 | -0.30848 | 96.22±0.55 |
| 42.5 | -0.06186 | 98.57±0.30 | -0.32389 | 96.65±0.54 |
| 45.0 | -0.06421 | 98.42±0.31 | -0.30355 | 96.89±0.08 |
| 47.5 | -0.06254 | 98.56±0.13 | -0.30862 | 96.81±0.31 |
| 50.0 | -0.07099 | 98.75±0.01 | -0.30195 | 96.27± 0.00 |

Fig. 23). See Table 8 for details of the spuriousness measure $M_{sp}$ and the worst-group accuracy. After performing k-means clustering, we use the estimation module to calculate the PID values.

All experiments are executed on NVIDIA RTX A4500. For the Waterbird dataset, the dimensionality reduction step takes around 50 minutes each for the background and foreground, and for the PID estimation with Liang et al. (2023) it takes $< 2$ seconds, and with James et al. (2018) takes around 50 seconds. For CMNIST, the whole process takes around $7-20$ seconds for the binary task and $15-50$ seconds for multiclass classification. Note that for the CMNIST dataset, we do not use an autoencoder for dimensionality reduction; rather, we directly perform k-means clustering followed by PID estimation due to the simplistic nature of this dataset.

## D.5 PID Estimator Analysis

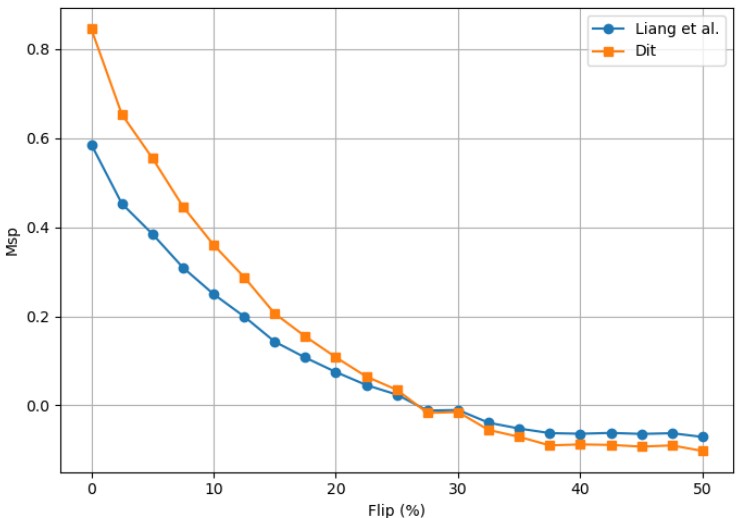

Figure 24: $M_{sp}$ for different PID estimators versus flip probability (%)

Fig. 24 presents a systematic comparison of the two PID estimators for the binary classification task on the CMNIST dataset across multiple flip probabilities. We observe that the estimated spuriousness measure $M_{sp}$ is largely consistent across estimators, with both following the same trend. These results suggest that our spuriousness measure can tolerate variations in the choice of the PID estimator and support the reliability of our main findings.

