# OpenReview forum: "Towards Formalizing Spuriousness of Biased Datasets Using Partial Information Decomposition"
_TMLR — Accepted by TMLR_

### Review · Reviewer_L5Yk · 2025-06-28

**Summary Of Contributions:**

The authors present a novel information-theoretic framework for preemptively quantifying spuriousness in datasets using a technique known as Partial Information Decomposition (PID). The PID can be leveraged to decompose total information about target variable into four non negative quantities: unique information in core / spurious features, redundant, and synergestic information. Using these a systematic derivation of a spuriousness measure is provided. It quantifies spuriousness in the dataset so that one may take preemptive measures to avoid them during training/fine-tuning. A three-step procedure: segmentation, dimensionality reduction via autoencoders, and PID estimation is proposed to achieve this. A comprehensive evaluation is done across six benchmark datasets to show the utility and effectiveness of the proposed framework. The authors empirically observe a negative correlation between proposed spuriousness measure and the post-training worst-group accuracy.

**Audience:**

Yes

**Broader Impact Concerns:**

I do not see any obvious potential concerns.

**Claims And Evidence:**

Yes

**Requested Changes:**

Please check the weaknesses section above.

**Strengths And Weaknesses:**

Strengths:
1. I like the idea of trying to quantify spuriousness of a dataset. While previous works just provide a handwavy idea of what is "spurious" the authors here try to systematically build on the notion of spuriousness and try to measure it, and also show how it correlates with accuracy measures.
2. The many datasets used in the empirical evaluation shows the reliability and consistency and effectiveness of the method.
3. Well written and easy to understand paper.
4. The connection they make to statistical decision theory is very interesting!

Weakness:
1. This framework definitely depends on apriori knowing or defining what constitutes core versus spurious features, which may limit the applicability in scenarios where such distinctions are not known apriori. Please throw a little more light on such scenarios and what the authors suggest or propose in those cases, maybe I didn't fully understand the scope of this paper. I'm also concerned about introducing subjective bias in feature categorization; the authors can suggest a few things in this regard.
2. The dimensionality reduction and clustering techniques surely may lead to loss of critical information. I want to know how this affects the reliability of PD estimates, some suggestions/solutions to overcome this, and a sensitivity analysis with-respect-to the discretization parameters is really helpful I feel.
3. The method overall is feasible, but I want to know the computational complexity of the algorithm, and the procedure involved, particularly the iterative autoencoder training, and the convex optimization protocol for the PID estimation. Maybe this is in the supplementary and I missed it, if so please point me to it.
4. Please include statistical tests for the correlations reported than just visual trends that are just observational.
5. Please release the code or point me to a github link. Maybe I missed it in the paper. That'll clarify a lot of other minor doubts I have about the methodology and implementation.
6. It would have also been very nice if there would have been some theoretical analysis connecting the PID measure to generalization bounds, would have greatly strengthened the claim. Have the authors thought about this and have any suggestions ideas in this regard.

Despite these weaknesses, I still feel the positives of the paper outweight the negatives, and I'm happy to accept the paper only if my questions and concerns raised above are convincingly addressed.

---

> ### Author Response · Authors · 2025-08-21
> **Response to Reviewer L5Yk: Part 1**
>
> We thank the reviewer for their review and for taking the time to share their thoughts and comments. We are glad that the reviewer appreciated our contribution: "I like the idea of trying to quantify spuriousness of a dataset. While previous works just provide a handwavy idea of what is "spurious" the authors here try to systematically build on the notion of spuriousness and try to measure it, and also show how it correlates with accuracy measures."
>
> We have updated our paper with additional experiments on 9 different variants of the Waterbirds dataset and 21 flip probabilities of the CMNIST dataset. These updates include statistical tests for the correlations (Section 5: Fig.7 and Fig. 14), a detailed sensitivity analysis w.r.t. the discretization parameter (Appendix E3, Additional Details on Clustering), and a computational complexity analysis (Appendix E.4.4). For the reviewers' convenience, we have highlighted the edited text, figure captions, and tables in blue in the revised manuscript.
>
> **Weakness 1: On applicability and subjective bias in
> feature categorization**
>
> We acknowledge the concern that identifying a priori what constitutes core versus spurious features can be challenging, and in some cases may introduce subjective bias. We have now further highlighted this in Appendix B (Limitations). This is indeed a valid limitation that is an active area of research and applies not only to our framework, but also to many existing approaches in the domain of robustness and spurious correlation analysis, e.g., [1].
>
> However, our method is designed to provide preemptive insights into spuriousness, even in settings where the distinction between core and spurious features is not explicitly defined. As demonstrated in Section 5 (Experimental Results), for datasets like **Spawrious** where foreground segmentation masks are not readily available, semantic segmentation models can be employed to generate these masks, enabling the separation of core and spurious features. We also employ *automatic segmentation using CLIPSeg [3]* (see Section 5, 3. Segmentation to obtain features), which isolates the foreground objects from the background without relying on pre-defined training classes or explicit labels for spurious features.
>
> Moreover, our framework has been successfully applied to **tabular datasets** such as the Adult dataset, where the segmentation module is not involved. This highlights its broader applicability beyond the tasks where features are spatially separable. When spuriousness arises from non-spatial factors (e.g., color or texture biases or gender), as in datasets such as **Colored MNIST and CelebA**, our method remains effective by leveraging partial knowledge about the core and spurious features. *Some works also mention that without any additional information or assumptions about the core or spurious features, it is generally not possible to address this problem as mentioned in [1].*
>
> We also emphasize that the subjective bias inherent in defining core or spurious features is a broader issue in the field. Recent works have tried to mitigate this by inferring environments or contexts automatically — such as the use of proxy models [1] or prompting techniques in MLLMs [2] to isolate and interpret potential spurious cues. In future work, we can incorporate our framework to these settings where we can utilize the partial knowledge of spurious features as well.
>
> In conclusion, our approach remains effective even with minimal prior knowledge of spurious features, which makes it adaptable to a wide range of settings, including image and tabular data with spatial or non-spatial biases. Our approach provides a useful preemptive diagnostic tool that can help identify potential biases in datasets prior to model training, offering a valuable addition to existing methods.
>
> References:
>
> [1] Yongqiang Chen, Yatao Bian, Kaiwen Zhou. Does Invariant Graph Learning via Environment Augmentation Learn Invariance?
>
> [2] Shenyu Lu et al. Mitigating Spurious Correlations in Zeroshot Multimodal Models.
>
> [3] Timo Lüddecke and Alexander Ecker. Image segmentation using text and image prompts.

---

> > ### Author Response · Authors · 2025-08-21
> > **Response to Reviewer L5Yk: Part 2**
> >
> > **Weakness 2: On dimensionality reduction and clustering**
> >
> > We acknowledge that dimensionality reduction and clustering can potentially result in the loss of critical information. Notably, existing approaches to compute Partial Information Decomposition (PID) measures struggle to handle continuous high-dimensional data effectively. One technique [1] works for 1-D random variables and another technique [2] (Gaussian PID) assumes Gaussianity on the random variables. In this situation, dimensionality reduction and discretization is only natural (and also done in several prior works in information theory). For a given number of clusters, we observe consistent trends in the spuriousness measure $M_{sp}$ across different variations of the dataset even if individual PID values vary slightly. Specifically, we compute PID values for cluster counts of 5, 10, and 20 for Waterbird dataset. As shown in Table 3 Appendix E.3, the PID components — and thus $M_{sp}$ — are sensitive to the number of clusters. Nevertheless, when moving from the unbalanced to the class-balanced setting, the spuriousness measure $M_{sp}$ consistently decreases across all cluster counts as expected, indicating that essential information is retained despite dimensionality reduction. A similar sensitivity analysis on the CMNIST dataset (Appendix E.3, Fig. 16) confirms this pattern. We observe that for CMNIST with two target classes, the computational time increases significantly — from approximately 6.8 seconds (cluster size 5) to 470.63 seconds (cluster size 50) — when the flip rate is set to 0\% (see the table below). But, most importantly, we observe that for more than 10 clusters, the optimization for PID computation sometimes fails to converge even after 50,000 iterations. Based on these observations and inspiration from previous work [3], we choose the number of clusters to be 10, striking as a trade-off between preserving sufficient information and ensuring computational efficiency and faster convergence.
> >
> > ### Table 1: PID estimation time with varying number of clusters for CMNIST (flip rate 0\%)
> > | Number of clusters | Time (seconds) |
> > |--------------------|----------------|
> > | 5                  | 6.8            |
> > | 10                 | 18.69          |
> > | 20                 | 65.12          |
> > | 30                 | 153.29         |
> > | 50                 | 470.63         |
> >
> >
> >
> > References:
> >
> > [1] Ari Pakman et al. Estimating the unique information of continuous variables. Advances in neural information processing systems, 34:20295–20307, 2021.
> >
> > [2] Praveen Venkatesh et al. Gaussian partial information decomposition: Bias correction and application to high-dimensional data. Advances in Neural Information Processing Systems, 36,
> > 2024.
> >
> > [3] Xifeng Guo, Xinwang Liu, En Zhu, and Jianping Yin. Deep clustering with convolutional autoencoders.
> >
> > **Weakness 3: On computational complexity**
> >
> > Our framework has two key components: dimensionality reduction and PID estimation. For dimensionality reduction, we use a self-supervised learning-based approach, training a deep learning model — specifically, an autoencoder — that jointly performs clustering and dimensionality reduction. The PID estimation is then performed using an iterative optimization-based method, which relies on repeated cycles to obtain an optimal solution for the PID terms.
> >
> > The autoencoder training for the Waterbird dataset takes approximately 50 minutes each for the foreground and background while the convex optimization protocol for the PID estimation with [1] takes less than 2 seconds and with [2] takes around 50 seconds. Our experiments on the CMNIST dataset show that for more than 10 clusters, the optimization for PID calculation sometimes fails to converge even after 50,000 iterations. Therefore, we conduct all our experiments using 10 clusters.
> > We have now included a discussion of the computational complexity in Appendix E.4.4, covering both the Waterbird dataset (the most computationally intensive case) and the CMNIST dataset (the least expensive). We will be happy to include runtime details for the other datasets and settings upon request or in future revisions for completeness.
> >
> > References:
> >
> > [1] Paul Pu Liang et al. Quantifying and modeling multimodal interactions: An information decomposition framework.
> >
> > [2] R. G. James, C. J. Ellison, and J. P. Crutchfield. dit: A Python package for discrete information theory.

---

> > > ### Author Response · Authors · 2025-08-21
> > > **Response to Reviewer L5Yk: Part 3**
> > >
> > > **Weakness 4: On statistical tests correlating spuriousness measure $M_{sp}$ and worst-group accuracy**
> > >
> > > We agree that visual trends alone may not suffice. Hence, we have now calculated the Pearson correlation coefficient (r) with corresponding p-value between spuriousness measure $M_{sp}$ and worst-group accuracy for the Waterbird and CMNIST datasets (see Fig. 7 and Fig. 14 and also the following table). To ensure that the statistical tests are meaningful, we include a broader range of variations for the corresponding datasets. Specifically, we consider 21 flip rates, resulting in 21 distinct variations of the CMNIST dataset, and a total of 9 variations of the Waterbird dataset (see Table 8 in Appendix E.4.4 and Appendix E.4.1 for details). These tests confirm that the observed trends are statistically significant and are not due to random variation.
> > >
> > > ### Table 2: Correlation analysis between spuriousness measure $M_{sp}$ and worst-group accuracy
> > >
> > > | Datasets        | Correlation Coeff. (r) | p-value  |
> > > |-----------------|-------------------------|----------|
> > > | Waterbird       | -0.89                   | 0.001470 |
> > > | CMNIST (Binary) | -0.72                   | 0.000260 |
> > > | CMNIST (Multi)  | -0.68                   | 0.000645 |
> > >
> > > **Weakness 5: On the link to implementation**
> > >
> > > We appreciate the interest of the reviewer in the implementation. An anonymous link to the code for better understanding of our methodology is provided here: [https://drive.google.com/drive/folders/1GP5METUeYh321uVNBLiaXbt2ICLRK_L0?usp=sharing](https://drive.google.com/drive/folders/1GP5METUeYh321uVNBLiaXbt2ICLRK_L0?usp=sharing)
> > >
> > > We will make the code publicly available on GitHub upon acceptance.
> > >
> > > **Weakness 6: On theoretical analysis connecting $M_{sp}$ to generalization bounds**
> > >
> > > We appreciate the reviewer’s insightful suggestion regarding a theoretical analysis connecting the spuriousness measure $M_{sp}$ to generalization bounds. While our current work primarily focuses on the empirical validation of the measure of spuriousness $M_{sp}$, we agree that establishing a formal theoretical connection to generalization guarantees would significantly enhance the framework. Recent work, such as [1], has explored information-theoretic approaches to derive tight bounds on the performance of Bayes' optimal classifier $P_{acc}(f*)$ as a function of total information $I_p(X_1, X_2; Y)$ in multimodal settings. In our context, the input features $X_1$ and $X_2$ can be interpreted as representations of the foreground and background, respectively. Since total information decomposes into redundant, unique, and synergistic components, it follows that $P_{acc}(f*)$ can, in principle, be bounded in terms of these PID quantities. We would explore the intriguing possibility of obtaining similar bounds using our proposed spuriousness measures, which is a promising direction for future work.
> > >
> > > References:
> > >
> > > [1] Paul Pu Liang et al. Multimodal learning without labeled multimodal data: Guarantees and applications. International Conference on Learning Representations (ICLR), 2024.

---

> > > > ### Comment · Reviewer_L5Yk · 2025-08-29
> > > > **I'm happy with the edits and changes**
> > > >
> > > > Dear Authors, I've read through your response and changes, and am decently satisfied. I'm willing to upgrade my score, leaning towards a weak accept at this point.

---

### Review · Reviewer_4T6E · 2025-07-17

**Summary Of Contributions:**

Spurious correlations are non-causal associations in datasets that mislead machine learning models, degrading generalization. The paper introduces a new framework based on Partial Information Decomposition (PID) to quantify such spurious correlations in datasets before the training of the models. The spurious disentangler is the key innovation, where core features (e.g. foreground) are separated by spurious ones (background) by leveraging an auto encoder architecture, and are then further analysed by PID.

The method in this paper is a novel contribution towards addressing spurious correlations, with potential to improve the reliability and fairness of machine learning models across diverse applications.

**Audience:**

Yes

**Broader Impact Concerns:**

No concerns.

**Claims And Evidence:**

Yes

**Requested Changes:**

Please revise the structure of the paper. I suggest shortening the introduction, renaming Main Results into Material and methods, and moving in the methods the description of the models, data and compute used for the experiments to make the paper easier to follow than its current version.

**Strengths And Weaknesses:**

The method is novel and grounded in solid information-theoretic principles.

Strengths:
- The use of PID is innovative and well grounded in information theory, which I really appreciate.
- The experimental results are strong and supportive of the theoretical claims, being compared across six benchmark datasets.

Weaknesses:
- I find the paper structure very confusing. While I appreciate the preliminaries section being grounded in the theoretical framework for the method, there is  no real Method section to describe the algorithm step by step, which I find very confusing to follow. Even worse, the architecture is presented in the Main Results section, which I find even more confusing. Besides, a lot of details about the overall framework are given in the introduction, which caused me to go back and forth through the paper to understand the methods.
- From my understanding, the framework requires to know the separation between core and spurious features beforehand. However, this may not always be possible to provide as an input.

---

> ### Author Response · Authors · 2025-08-21
> **Response to Reviewer 4T6E**
>
> We sincerely thank the reviewer for their positive feedback and for recognizing the strengths of our work. We appreciate the comment: "The use of PID is innovative and well grounded in information theory, which I really appreciate." For the reviewers' convenience, we have highlighted the edited text, figure captions, and tables in blue in the revised version.
>
> **On paper structure**
>
> We thank the reviewer for valuable feedback regarding the organization of the paper. In the revised version, we have reorganized the content to improve clarity. A dedicated "Methodology" section has been added, where the proposed algorithm is now described step by step, and also presented in a formal algorithm environment (see Algorithm 1). We put our theoretical findings in a separate section titled "Theoretical Contributions". Additionally, we have streamlined the "Introduction" by reducing detailed descriptions to avoid redundancy. We remain open to incorporating further modifications if suggested.
>
> **On the separation between core and spurious features**
>
> We acknowledge the reviewer’s concern that our framework assumes a prior separation of core and spurious features, which may not always be feasible in practice. This is indeed a valid limitation that is an active area of research and applies not only to our framework, but also to many existing approaches in the domain of robustness and spurious correlation analysis, e.g., [2][3]. We further highlight this aspect in our limitations (see Appendix B, Limitations). Nevertheless, our strategy is applicable in several use-cases as a measure of dataset quality evaluation/explainability prior to expensive model training (without knowing the exact split of the core/spurious features apriori).
>
> As we show in Experimental results (Section 5), for spatially separable core and spurious features, we can perform segmentation. For example, for datasets like **Spawrious** where foreground segmentation masks are not readily available, semantic segmentation models can be employed to generate these masks, enabling the separation of core and spurious features. One can also employ *automatic segmentation using CLIPSeg [1]* (see Section 5, 3. Segmentation to obtain features), which isolates the foreground objects from the background without relying on pre-defined training classes or explicit labels for spurious features.
>
> Our framework has been successfully applied to **tabular datasets** such as the Adult dataset, where the segmentation module is not involved. This highlights its broader applicability beyond the tasks where the features are spatially separable. When spuriousness arises from non-spatial factors (e.g., color or texture biases, gender), as in datasets such as **Colored MNIST and CelebA**, our method remains effective by leveraging partial knowledge about the core and spurious features. Some works also mention that without any additional information or assumptions about the core or spurious features, it is generally not possible to address this problem [2].
>
> In conclusion while disentangling causal and spurious features remains a significant challenge in OOD tasks, our method is still applicable to several use-cases for evaluating dataset quality without knowing the exact split of the core and spurious features apriori. Our approach provides a useful preemptive diagnostic tool that can help identify potential biases in datasets prior to model training, offering a valuable addition to existing methods.
>
> References:
>
> [1] Timo Lüddecke and Alexander Ecker. Image segmentation using text and image prompts.
>
> [2] Yongqiang Chen, Yatao Bian, Kaiwen Zhou. Does Invariant Graph Learning via Environment Augmentation Learn Invariance?
>
> [3] Martin Arjovsky, Léon Bottou, Ishaan Gulrajani, and David Lopez-Paz. Invariant risk minimization.

---

### Review · Reviewer_rHRs · 2025-08-14

**Summary Of Contributions:**

The authors propose an information-theoretic framework for preemptively quantifying dataset spuriousness by using Partial Information Decomposition to break down predictive information into unique, redundant, and synergistic components. They introduce a spuriousness measure that captures the tendency of models to rely on spurious rather than core features, and present the Spuriousness Disentangler, an autoencoder-based pipeline combining segmentation, dimensionality reduction, and joint distribution estimation to compute these components for high-dimensional data. Experiments on six benchmark datasets under varied bias conditions show a consistent negative correlation between the proposed measure and post-training worst-group accuracy, supported by visual evidence from Grad-CAM and IoU analyses, demonstrating the method’s effectiveness for dataset quality assessment and interpretability.

**Audience:**

Yes

**Claims And Evidence:**

Yes

**Requested Changes:**

I wish the authors would address or give more discussion on the mentioned weaknesses.

**Strengths And Weaknesses:**

Strengths:
- The method builds on Partial Information Decomposition. It connects this with Blackwell sufficiency from decision theory. This gives a structured way to separate unique, redundant, and synergistic information. It also explains when a feature is indispensable or interchangeable.

- The spuriousness metric is carefully derived. The authors use theory and counterexamples to rule out weaker alternatives like plain mutual information. The final measure captures when spurious features dominate, when core features dominate, and when both act together.

- The framework is tested on six datasets with varied bias types. Results consistently show a negative link between the measure and worst-group accuracy. Visual tools like Grad-CAM and IoU confirm whether the model focuses on the right features. This strengthens trust in the results.

Weaknesses:
- Despite being mentioned by the authors, I am still worried about the PID estimation being highly data-dependent. Small sampling changes or noise in the dataset can shift the values a lot. This makes results less stable for small datasets or those with high variability.
- The pipeline depends on clustering latent features, then estimating PID on a discretized joint. The spuriousness score moves with the number of clusters, runtime grows quickly, and the PID optimizer sometimes fails to converge. This makes the method brittle to design choices and scale.
- The method fixes a particular PID estimator for most experiments. It switches to a different estimator for multiclass. Different valid PID formalisms can partition information differently so that the score may inherit estimator-specific behavior. The paper does not study this dependence.

---

> ### Author Response · Authors · 2025-08-21
> **Response to Reviewer rHRs: Part 1**
>
> We thank the reviewer for their thoughtful feedback and for highlighting the strengths of our work.
>
> **On data dependency of PID estimation**
>
> We agree that the data-dependent nature of PID estimation can sometimes make individual estimates of unique, redundant, or synergistic information sensitive to even small changes in the dataset. We have now further elaborated this in Appendix B (Limitations). In fact, estimating information-theoretic quantities such as mutual information is inherently difficult in high-dimensional settings [7] including formal limits on sample complexity for all estimators [8], and PID estimation would also inherit these challenges. Moreover, PID requires solving an additional optimization problem beyond mutual information computation, which can introduce additional difficulties. Existing methods for computing PID measures are limited in scope: for instance, one approach [1] is restricted to one-dimensional random variables, while another [2] (Gaussian PID) relies on the assumption that the random variables are Gaussian. [3] proposes a method to compute PID values for high-dimensional continuous data by first applying discretization and dimensionality reduction, followed by PID computation by solving a convex optimization. Our **novelty** lies in joint dimensionality reduction and clustering (discretization) to alleviate the information loss, followed by PID computation.
>
> Although estimating PID values is subject to notable limitations, the promise of PID is considerable, as demonstrated by their successful applications in diverse machine learning challenges [4][5][6]. **In our work, we also observe a consistent negative correlation between the proposed spuriousness measure and worst-group accuracy (validated with statistical tests), indicating that the proposed PID-based spuriousness measure still remains informative across different estimation techniques.** Our framework offers a preemptive way to assess dataset quality, which can substantially reduce computational cost as long as the measures have similar expected trends even if the exact estimate varies slightly (also see the newly added Appendix E.5, PID Estimator Analysis).
>
> In conclusion, despite the inherent challenges of estimation, PID-based analyses and information-theoretic measures, in general, provide a powerful tool for understanding dataset properties and guiding more explainable and trustworthy machine learning frameworks. Studying alternate estimation techniques will be an interesting direction of future research.
>
> References:
>
> [1] Ari Pakman et al. Estimating the unique information of continuous variables. Advances in
> neural information processing systems, 34:20295–20307, 2021.
>
> [2] Praveen Venkatesh et al. Gaussian partial information decomposition: Bias correction
> and application to high-dimensional data. Advances in Neural Information Processing Systems, 36, 2024.
>
> [3] Paul Pu Liang et al. Quantifying \& modeling multimodal interactions: An information decomposition framework. Advances in Neural Information Processing Systems 36 (2023): 27351-27393.
>
> [4] Hamman, Faisal, and Sanghamitra Dutta. Demystifying local and global fairness trade-offs in federated learning using partial information decomposition. Federated Learning and Analytics in Practice: Algorithms, Systems, Applications, and Opportunities. 2023.
>
> [5] Paul Pu Liang et al. Multimodal learning without labeled multimodal data: Guarantees and applications. International Conference on Learning Representations (ICLR), 2024.
>
> [6] Shaurya Dewan et al. DiffusionPID: Interpreting Diffusion via Partial Information Decomposition. Advances in Neural Information Processing Systems 37 (2024): 2045-2079.
>
> [7] Belghazi, Mohamed Ishmael, et al. Mutual information neural estimation. International conference on machine learning. PMLR, 2018.
>
> [8] McAllester, David, and Karl Stratos. Formal limitations on the measurement of mutual information. International Conference on Artificial Intelligence and Statistics. PMLR, 2020.

---

> > ### Author Response · Authors · 2025-08-21
> > **Response to Reviewer rHRs: Part 2**
> >
> > **On stability of the pipeline across design choices**
> >
> > We thank the reviewer for this observation. The pipeline indeed relies on clustering latent features before computing PID on a discretized joint distribution, which can introduce sensitivity to design choices such as the number of clusters. More clusters would lead to less loss of information during estimation but too many clusters can also affect the complexity and convergence of the PID computation. We conducted a sensitivity analysis varying the number of clusters and observed that the trend between the spuriousness measure $M_{sp}$ and worst-group accuracy remains consistent (see Appendix E.3, Additional Details on Clustering). Based on these experiments, we found that using 10 clusters provides a good choice across the datasets, balancing stability, computational efficiency, and informativeness of the spuriousness measure. We note that once the number of clusters, e.g., 10 is fixed, the second step of PID computation by solving the optimization is of similar complexity across datasets since it is an optimization over three random variables, two taking 10 values and the target $Y$ taking 2 values (binary classification). Other related works, e.g., [1] have also found 10 clusters to be a good hyperparameter choice though the estimation strategy is different.
> >
> > References:
> >
> > [1] Xifeng Guo, Xinwang Liu, En Zhu, and Jianping Yin. Deep clustering with convolutional autoencoders. International conference on neural information processing. Cham: Springer International Publishing, 2017.
> >
> >
> > **On PID estimators**
> >
> > We acknowledge that different PID formalisms can partition information differently, which may lead to estimator-specific effects on the spuriousness measure. In the revised version, we include a systematic study of the two estimators for the binary classification task on the CMNIST dataset across multiple flip probabilities (see Appendix E.5, PID Estimator Analysis). We observe that the estimated spuriousness measure $M_{sp}$ is largely consistent across estimators, with both following the same trend. Our approach will still be informative as long as it has similar expected trends even if the exact PID estimates vary slightly. We use [1] to estimate PIDs for the multiclass CMNIST dataset, as it provides faster convergence and improved stability. These results suggest that our spuriousness measure can still tolerate different PID estimators and support the reliability of our main findings.
> >
> > References:
> >
> > [1] Paul Pu Liang et al. Quantifying \& modeling multimodal interactions: An information decomposition framework. Advances in Neural Information Processing Systems 36 (2023): 27351-27393.

---

### Decision · Action_Editor_skmb · 2025-10-03

**Recommendation:** Accept as is

**Additional Comments:**

The paper tackles the important problem of characterizing spuriousness of a dataset. While many papers propose methods to avoid learning spurious features, the definition is largely hand-wavy. I appreciate the authors' attempt to define spuriousness and provide a metric for a given dataset. Reviewers noted the strengths: information theory justification and empirical experiments across multiple benchmarks showing that the metric correlates with downstream metrics such as worst-group accuracy.

After the revision, the clarity of the paper has improved. There are still some limitations, such as dependence of some knowledge of the spurious features' characteristics, that may lessen broader application.
Still, as reviewers noted, the paper provides good justification and experiments for the proposed metric, which leads me to recommend Accept.

**Audience:**

Yes

**Audience Explanation:**

Yes, spurious features (and how to avoid learning them) has been a widely studied problem. This paper aims to characterize and measure spuriousness, which would be interest to many.

**Claims And Evidence:**

Yes

**Claims Explanation:**

The paper proposes a metric for spuriousness of a dataset. The metric is applied to six benchmark datasets and correlated with downstream metrics such as worst-group accuracy. The metric is both theoretically and empirically justified.